# On the Effectiveness of Lipschitz-Driven Rehearsal in Continual Learning

**Lorenzo Bonicelli**[1]    **Matteo Boschini**[1]    **Angelo Porrello**[1]
**Concetto Spampinato**[2]    **Simone Calderara**[1]

[1]AImageLab - University of Modena and Reggio Emilia
[2]PeRCeiVe Lab - University of Catania

## Abstract

Rehearsal approaches enjoy immense popularity with Continual Learning (CL) practitioners. These methods collect samples from previously encountered data distributions in a small memory buffer; subsequently, they repeatedly optimize on the latter to prevent catastrophic forgetting. This work draws attention to a hidden pitfall of this widespread practice: repeated optimization on a small pool of data inevitably leads to tight and unstable decision boundaries, which are a major hindrance to generalization. To address this issue, we propose Lipschitz-DrivEn Rehearsal (LiDER), a surrogate objective that induces smoothness in the backbone network by constraining its layer-wise Lipschitz constants w.r.t. replay examples. By means of extensive experiments, we show that applying LiDER delivers a stable performance gain to several state-of-the-art rehearsal CL methods across multiple datasets, both in the presence and absence of pre-training. Through additional ablative experiments, we highlight peculiar aspects of buffer overfitting in CL and better characterize the effect produced by LiDER. Code is available at https://github.com/aimagelab/LiDER.

## 1  Introduction

The last few years have seen a renewed interest in aiding Deep Neural Networks (DNNs) to acquire new knowledge and, at the same time, retain high performance on previously encountered data. In this regard, the mitigation of *catastrophic forgetting* [52] has driven the recent research towards novel incremental methods [42, 68], often framed under the field of Continual Learning (CL). Among other valid CL strategies, *rehearsal approaches* caught the attention of a large body of literature [62, 21, 15] thanks to their advantages. Simply, they maintain a small fixed-size buffer containing a fraction of examples from previous tasks; afterward, these examples are mixed together with the ones of the current task, hence provided continuously as training data. In this respect, different approaches establish different regularization strategies on top of the retained examples [63, 49], as well as which kind of information to store (*e.g.* model responses [15, 12], explanations [25], etc.).

In spite of their widespread application, these approaches fall into a common pitfall: as the memory buffer holds only a small fraction of past examples, there is a high risk of overfitting on that memory [75], thus harming generalization. Several approaches mitigate such an issue through data-augmentation techniques, either by generating different versions of the same buffer datapoint [7, 15] or by combining different examples into a single one [14, 11]. Other works [3, 4, 16, 85], instead, select carefully the valuable samples that should be inserted into the buffer: they argue that random selection may pick non-informative and noisy instances, affecting the model generalization.

This work tackles the issue described above from a different perspective, viewing *catastrophic forgetting* in the light of **the progressive deterioration of decision boundaries** between classes.

36th Conference on Neural Information Processing Systems (NeurIPS 2022).

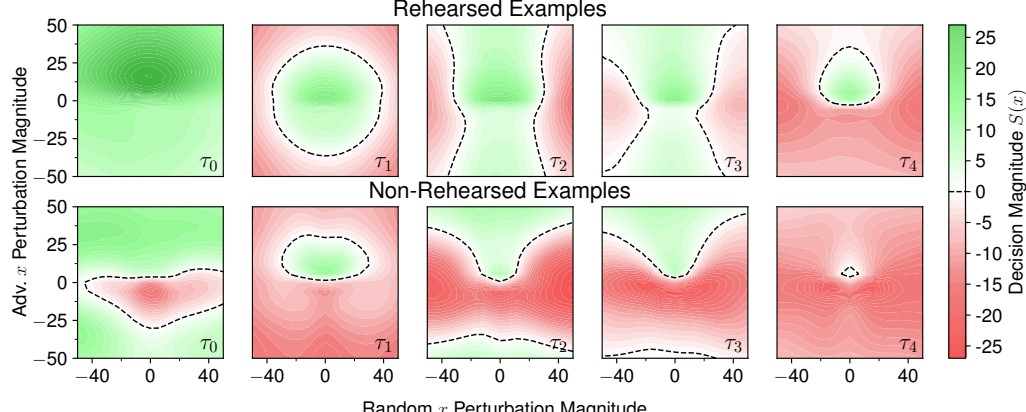

Figure 1: Diagrams derived from [87] describing how the magnitude of an input perturbation affects the model's prediction, measured as the difference between the correct logit and the maximum one. The dashed lines delimit the green areas, where the correct response is preserved. The analysis is carried out on the examples of the first task of Split CIFAR-10 and spans across tasks progressing, from $\tau_0$ (left) to $\tau_4$ (right). In different rows, the decision surfaces around datapoints either contained in the memory buffer (first) or non-rehearsed (second). We refer to Sec. 5.3 for additional insights.

Indeed, while for the examples of the current task we expect the decision boundaries to be already smooth and robust against local perturbations, the same could not be said for past examples. In fact, the restrained access to only a small portion of past tasks increases the epistemic uncertainty [38] of the model: as a consequence, we expect the decision surfaces tied to past classes to slowly erode everywhere, with the exception of certain input regions *i.e.*, those close to the neighborhood of buffer datapoints (thanks to their repeated optimization). We refer to Fig.1 for a visualization of such phenomenon, which shows the evolution of the decision surfaces around the points of the first task of Split CIFAR-10, from the first (*left*) to the last one (*right*). We differentiate the target of the analysis (rehearsed examples *vs.* non-rehearsed examples) in distinct rows: as can be seen, the green area – the input region where the model outputs correct predictions against local input perturbations [87]) – tightens around buffer datapoints (*first row*) and erodes for non-rehearsed examples (*second row*).

Such an intuition motivates our research of novel mechanisms for guaranteeing the robustness of the decision boundaries. To this aim, we resort to enforcing the Lipschitz continuity of the model w.r.t. its input: indeed, a long-standing research trend [82, 72, 45, 46, 80, 29] has pointed out that such a property favors generalization capabilities and robustness to adversarial attacks. In particular, constraining the Lipschitz constant of a model – intuitively, a bound on how much the model's response can change in proportion to a change in its input [5] – has proven to strengthen the decision surface around a point [22, 45, 46, 80, 91], preventing attacks of a given magnitude from changing the output of the classifier.

While these works assessed Lipschitz regularization in the classical scenario (*i.e.*, single joint i.i.d. task), we advocate that it is even more beneficial in continual learning, particularly for those approaches based on replay memories. As shown in the inset figure, without explicit regularization, the Lipschitz constant of a model increases for smaller memory buffers: in other terms, its corresponding function space becomes increasingly sensitive

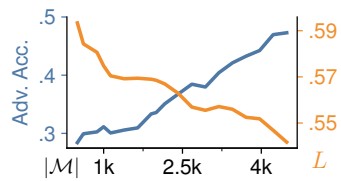

w.r.t. local input perturbations (as also highlighted by the lower accuracy attained in the presence of adversarial attacks). In light of the above considerations, we ascribe such a tendency to the higher uncertainty, which derives from subjecting the model to a low-data training regime.

To the best of our knowledge, our work is the first attempt to assess the effectiveness of Lipschitz-constrained DNNs in continual learning: in particular, we have equipped several widely-known and state-of-the-art rehearsal approaches with our Lipschitz-guided optimization objective named **Lipschitz-DrivEn Rehearsal (LiDER)**, showing that it systematically leads to better results in several benchmarks.

## 2 Related works

**Continual Learning**. CL examines the capability of a deep model to learn from a sequence of non-i.i.d. classification tasks [23, 58] while preventing the onset of *catastrophic forgetting* [52]. To achieve this goal, models are trained according to specifically designed strategies, meant to influence their evolution and maximize the retention of previously acquired knowledge.

Among them, *regularization methods* work by introducing additional constraints in the form of loss terms; they are designed to limit the amount of total change either in parameter space [42, 89, 19, 2] or functional space [47, 10]. Differently, *structural methods* purposefully organize the allocation of model capacity to prevent interference and facilitate parameter sharing [1, 51, 66, 37]. Lastly, *rehearsal methods* store and reuse a subset of previously seen data-points to prevent overfitting on current data and avert forgetting [16, 49, 4, 3]. While *rehearsal* strategies are by far the most frequently adopted thanks to their effectiveness and flexibility [26, 4], it is not uncommon to adopt solutions combining multiple approaches [32, 1, 15, 60, 18]. In this paper, we similarly propose a strategy that leverages an existing replay memory buffer to compute an additional regularization term, aimed at conditioning the learning dynamics and avoiding overfitting.

CL evaluations are often carried out in the so-called Task-Incremental setting (TIL) [42, 89, 49, 23, 19] – that is, the model is provided a *task identifier* at test time to avoid interference across predictions of distinct tasks. However, recent works put an increasing focus on the harder Class-Incremental setting (CIL) [4, 32, 15, 81], which entails the production of a unified prediction encompassing all seen classes. W.r.t. the latter, TIL has been criticized as a less challenging and realistic benchmark [26, 74, 4]; we therefore conduct our main experiments on state-of-the-art CL models in the CIL setting[1].

**Lipschitz-based Regularization**. Naturally trained DNNs typically suffer from their overparametrization [90, 57], leading to the tendency to overfit the training data by producing jagged decision boundaries that closely fit the seen examples. On the contrary, a model's reliability depends on its capability for generalization, which is linked to the appearance of smooth decision boundaries [82, 8, 86, 29]. Starting from the first studies focusing on this simple dichotomy, the Lipschitz constant $L$ of a DNN has been established as a commonplace measure of both smoothness and generalization [82, 72, 39] and still constitutes a key ingredient for current evaluations of model capacity [8, 30].

Most notably, *Szegedy et al.* [72] verify that constraining $L$ reduces the model's vulnerability to adversarial perturbations. Many current approaches to Adversarial Learning similarly operate either by minimizing $L$ at the *global* or *local* level [46, 73, 45] or by devising models characterized by a small $L$ by design [22, 33]. In other areas, the smoothing effect of $L$-based regularization has been favorably applied to both GAN training [55] and neural fields [48].

## 3 Method

A CL problem usually involves learning a function $f$ from a stream of data, which we formalize as a succession of separate datasets $T = \{\tau_0, \tau_1, \ldots, \tau_{|T|}\}$, where $\tau_t = \{(\mathbf{x_i}, y_i)\}_{i=1}^{N_t}$ and $\tau_i \cap \tau_j = \varnothing$; the label set $Y_t$ for each $\tau_t$ are non-overlapping. In this setting, the ideal objective consists in minimizing the overall loss over all tasks experienced, formally:

$$f^* = \operatorname*{argmin}_{f} \mathbb{E}_{t=0}^{|T|}\left[\mathbb{E}_{(\mathbf{x},y)\sim\tau_t}\left[\mathcal{L}(f(\mathbf{x}), y)\right]\right], \tag{1}$$

where $\mathcal{L}$ is an appropriate loss for solving the task at hand. In a continual scenario, only data from the current task $\tau_t$ is freely available; therefore, CL methods need to maintain knowledge from the past $t - 1$ tasks in order to solve the overall problem.

For the sake of simplicity, we consider a feed forward neural network $f(\cdot) = (H^K \circ \sigma^K \circ H^{K-1} \circ \sigma^{K-1} \circ \ldots H^1)(\cdot)$, *i.e.*, a sequence of $\sigma$-activated linear functions $H^k(\mathbf{h}) = \mathbf{W}_k^T \mathbf{h}$ (biases are omitted). A final projection head $g(\cdot) = \operatorname{softmax}(\cdot)$ is applied to produce per-class output probabilities. As stated in other works [29], other common transformations that make up DNNs (*e.g.*, convolutions, max-pool) can also be seen in terms of matrix multiplications, thus making our approach applicable to more complex networks.

---

[1]However, we remark that TIL can also be useful, as it reveals *forgetting* disentangled from other incremental learning effects. This motivates us to adopt TIL in some of our additional experiments.

**Lipschitz continuity**. A function $f$ is said to be *Lipschitz continuous* if there exists a value $L \in \mathbb{R}^+$ such that the following inequality holds:

$$||f(\mathbf{x}) - f(\mathbf{y})||_2 \leq L||\mathbf{x} - \mathbf{y}||_2, \quad \forall \mathbf{x}, \mathbf{y} \in \mathbb{R}^n. \tag{2}$$

If such a value exists, the smallest $L$ that satisfies the condition is usually referred to as the Lipschitz norm $||f||_L$. Therefore, for a single point $x \in \mathbb{R}^n$, we obtain:

$$||f||_L = \sup_{x \neq y; y \in \mathbb{R}^n} \frac{||f(\mathbf{x}) - f(\mathbf{y})||_2}{||\mathbf{x} - \mathbf{y}||_2}. \tag{3}$$

Unfortunately, computing the Lipschitz constant of even the most simple multi-layer perceptron is a NP-hard problem [77]. Therefore, several works relied on its estimation by computing reliable upper bounds. As stated in [86, 67], an effective way to bound the Lipschitz constant of $f(\cdot)$ is to compute the constants of each linear projection $H^k$ and then aggregate them. In more detail:

$$||H^k||_L = \sup_{x \neq y; y \in \mathbb{R}^n} \frac{||W^T\mathbf{x} - W^T\mathbf{y}||_2}{||\mathbf{x} - \mathbf{y}||_2} = \sup_{\xi \neq 0; \xi \in \mathbb{R}^n} \frac{||\mathbf{W}_k \xi||_2}{||\xi||_2} = \sigma_{\max}(\mathbf{W}_k), \tag{4}$$

where $\sigma_{\max}(\mathbf{W}_k)$ is the largest singular value of the weight matrix $\mathbf{W}_k$ (also know as its spectral norm $||\mathbf{W}_k||_{SN}$). To account for non-linear composite functions (*e.g.*, the residual building blocks of most convolutional architecture), we leverage the following inequality:

$$||g(z(\mathbf{x})) - g(z(\mathbf{y}))||_2 \leq ||g||_L ||z(\mathbf{x}) - z(\mathbf{y})||_2$$
$$\leq ||g||_L ||z||_L ||\mathbf{x} - \mathbf{y}||_2 \Rightarrow ||z \circ g||_L \leq ||g||_L ||z||_L,$$

where $\mathbf{g}(\cdot)$ and $\mathbf{z}(\cdot)$ are two Lipschitz continuous functions characterized by the constants $||g||_L$ and $||z||_L$. In the case of ReLU-activated networks (but the following result can be extended to other common non-linear functions), the forward pass through $\sigma^l\ l = 1, 2, \ldots, L$ can be re-arranged as a matrix multiplication by a diagonal matrix $D^l \in \mathbb{R}^{d_l \times d_{l+1}}$ whose diagonal elements equal either zero or one. Therefore, their corresponding Lipschitz constant $||\sigma^l||_L \leq 1$. On top of that, we can compute an upper bound on the Lipschitz constant of the entire network:

$$||f||_L \leq ||H^K||_L \cdot ||\sigma^K||_L \cdot \ldots \cdot ||H^1||_L \leq \prod_{k=1}^{K} ||H^k||_L = \prod_{k=1}^{K} ||\mathbf{W}^k||_{SN}. \tag{5}$$

**Computing the spectral norm of weights matrices**. The computation of $||\mathbf{W}_k||_{SN}$ can be done [55, 29] naively through the Singular Value Decomposition (SVD), yielding, among the others, the largest singular value. Such approach has been applied in recent works [55, 29]; however, for complex structures (*e.g.*, convolutions or entire residual blocks) the SVD decomposition is inaccessible. Hence, we rely on the approximation introduced in [67] and compute the largest eigenvalue $\lambda_1^k$ of the Transmitting Matrix $\mathbf{TM}^k$ (which represents a good proxy of $||\mathbf{W}^k||_{SN}^2$):

$$\mathbf{TM}^k \triangleq \left[ (\mathbf{F}^k)^T (\mathbf{F}^{k-1}) \right]^T \left[ (\mathbf{F}^k)^T (\mathbf{F}^{k-1}) \right], \tag{6}$$

where $\mathbf{F}^k \in \mathbb{R}^{B \times d_k}$ is the L2-normalized feature map produced by the $l$-th layer from a batch of $B$ samples. Finally, our approach exploits the power iteration method [56] to compute the largest eigenvalue of $\mathbf{TM}^k$, which is backpropagation-friendly.

## 3.1 Lipschitz-Driven Rehearsal

In a continual setting, a model is asked *i)* to be adaptable to incoming samples from the stream (plasticity), and *ii)* to be accurate on past tasks (stability). We seek to ensure a balance between these clashing objectives through the two following loss terms.

**Controlling Lipschitz-continuity**. To mitigate overfitting on buffer datapoints, we firstly impose that each layer behaves as a $c$-Lipschitz continuous function, for a given real positive target constant $c_k$:

$$\mathcal{L}_{\text{c-Lip}} = \frac{1}{K} \sum_{k=0}^{K} |\lambda_1^k - c_k|. \tag{7}$$

---

[2] We refer the reader to [67] for additional justifications for this step.

During the computation of each $\lambda_1^k$, we discard the activation maps incoming from the examples of the current task. Indeed, as we have access to the entire training set (and not a subset as holds for old tasks), additional regularization is not needed: the decision boundaries tied to the current task are less prone to the risk of over-adapting to certain points. Regarding the target constants $c_k$, we could fix them as hyperparameters of our learning objective (as done in [48]) and exploit them as a sort of budget assigned to each layer; however, we empirically observed that it is beneficial, instead, learning these targets by means of gradient descent (see Sec. 5.4), especially in a CL scenario where there is no access to the full data distribution. Indeed, these can be interpreted as additional learnable parameters, which represent the appropriate level of strictness each layer should be subjected to.

To avoid trivial solutions, we also encourage the estimated upper bounds to be as much as possible close to zero:

$$\mathcal{L}_{0\text{-Lip}} = \frac{1}{K} \sum_{k=0}^{K} |\lambda_1^k|. \tag{8}$$

Intuitively, when $\lambda_1^k \to 0$, the outputs of the corresponding $k$-th layer have low sensitivity to changes in its input. In our intentions, this could relieve continuous rehearsal from eroding the decision surface in a way that fits well only certain examples (*i.e.*, those retained in the memory buffer).

**Overall objective**. The overall objective of LiDER combines the two introduced loss terms; formally:

$$\mathcal{L}_{\text{LiDER}} = \alpha \mathcal{L}_{\text{c-Lip}} + \beta \mathcal{L}_{0\text{-Lip}}. \tag{9}$$

This objective can be plugged in almost any rehearsal approach. For such a reason, we keep it general and avoid reporting the common loss terms asking for accurate predictions, as their form depends on the specific choices made by each approach. Finally, we further remark that the introduced loss terms require minimal additional computation. Moreover, they do not need additional samples to be retained, besides those that are already present in the memory buffer.

## 3.2    Relation with other regularization approaches

At first glance, the regularization of our approach can be understood as a mean to enforce flat minima for each of the tasks, as advocated by Mirzadeh et al. [54] and Yin et al. [84]. We remark that they reason in **parameter space** and pursue flatness of the loss landscape w.r.t. weights: namely, they encourage the model to be robust when perturbations are applied to its **weights**. Differently, we seek models that are robust w.r.t. changes in **input space**. The two lines may exploit the same mathematical tools – such as the Hessian and Lipschitz continuity – but build upon orthogonal axes (weights *vs* input): we recommend to always assess which of the two is used as reference property.

In this respect, the bridge between the two objectives is worth-exploring and still open to debate [83, 71, 87, 36]. The authors of [83] reported that, theoretically, no correlation exists between the Hessian w.r.t. weights and the robustness of the model w.r.t. the input. Such a statement is corroborated by Fig. 1 of [87]: although a flat minimum is reached in parameter space, non-smooth variations appear in input space. However, the authors of [83] empirically found that models with higher Hessian spectrum w.r.t. weights are also more prone to adversarial attacks. A similar thesis has been argued by the authors of [87], while the third result reported in [36] seems to refute it. Furthermore, Sec. 5.3 of our paper investigated the opposite link and revealed that CL models trained to be robust w.r.t. input changes tend to attain flatter minima in parameter space.

## 4    Experiments

To assess our proposal, we perform a suite of experiments through the Mammoth framework [12, 17, 9, 6, 27, 13, 14, 50], an open-source codebase introduced in [15] for testing CL algorithms. In particular, we show that our method can be easily applied to state-of-the-art replay methods and enhance their performance in a wide variety of challenging settings and backbone architectures. Moreover, we show that our proposal remains rewarding and can improve the generalization capabilities of CL models even when a pre-trained model is employed. Such scenario is important for a twofold reason: i) as shown in [53], pre-training implicitly mitigates forgetting by widening the local minima found in function space, thus making the model more robust to input perturbations; additionally, ii) we accommodate for real-world scenarios where pre-training is usually involved as an initial step. Due to space constraints, we kindly refer the reader to the supplementary material for additional experimental details (*e.g.*, optimizer, hyperparameters, etc.).

## 4.1 Benchmarks

We focus our evaluation on the CIL setting and rely on commonly used and challenging image classification tasks. In each experiment, classes from the main dataset are split into separate and disjoint sets, which are then used sequentially to train the models herein described.

**Split CIFAR-100**. An initial evaluation is carried by splitting the $32 \times 32$ images from the 100 classes of CIFAR-100 [43] into 10 tasks. In detail, we run two tests: *i)* using a randomly initialized ResNet18 [31] – which constitutes a common scenario in literature [89, 62, 21, 16]; *ii)* adopting the same backbone but with its parameters pre-trained on Tiny ImageNet [70].

**Split *mini*ImageNet**. This setting is designed to assess models on longer sequences of tasks. It splits the 100 classes of *mini*ImageNet [76] – a subset of ImageNet, with images resized to $84 \times 84$ – into 20 consecutive tasks. For this test, we opt for an EfficientNet-B2 [69] backbone with no pre-train.

**Split CUB-200**. A final, more challenging benchmark involves classifying large-scale $224 \times 224$ images from the Caltech-UCSD Birds-200-2011 [79] dataset, organized in a stream of 10 20-fold classification tasks. This evaluation is designed to highlight the importance of protecting pre-trained weights from forgetting. Indeed, due to the limited size of the training set, competitive performance can only be achieved if each task can benefit from the initialization [20, 88]. As backbone network, we opt for the commonly available ResNet50 architecture pre-trained on the ImageNet dataset [24].

We report results in terms of the classification accuracy averaged across all tasks (Final Avg. Acc., FAA). We refer the reader to supplementary materials for the Final Forgetting [19] (FF). For further reference, we include the results of a model jointly trained on all classes – which represents an ideal non-CL upper bound – and of a model trained sequentially without countermeasures to forgetting.

## 4.2 Comparison with rehearsal approaches

**Benchmarked models**. We conduct experiments on multiple state-of-the-art rehearsal-based models. If not specified, all methods use reservoir sampling [78] to update the memory buffer.

- **Experience Replay with Asymmetric Cross-Entropy (ER-ACE)** [17]: it enhances standard ER by introducing a separate loss for datapoints coming either from the stream and the buffer.
- **Dark Experience Replay (DER++)** [15]: the authors exploit self-distillation [28] to take advantage of the previously learned knowledge. In addition to storing the one-hot labels, they enforce consistent responses through time by minimizing the L2 norm between current and past logits.
- **eXtended Dark Experience Replay (X-DER)** [12]: it advances DER++ by introducing several amendments. In particular: *i)* it updates the logits contained in the memory buffer by implanting novel secondary information made available for past classes; *ii)* it exploits supervised contrastive learning [40] to prepare the model for future tasks. Among the variants discussed in Sec. 5.2 of [12], we use the lightweight version called X-DER w/ RPC [59] to prepare future heads.
- **Incremental Classifier and Representational Learning (iCaRL)** [62]: it learns a representation suitable for a nearest-neighbor classification w.r.t. class prototypes stored in the buffer. Additionally, forgetting is prevented by distilling the responses of the model's snapshot at the previous task to both current and replay examples. The buffer is managed through the herding strategy.
- **Greedy Sampler and Dumb Learner (GDumb)** [61]: it trains the model only on buffer datapoints at the task boundaries. By doing so, the authors meant to challenge the improvements made on CL.

Results of our evaluation can be found in Tab. 1: across the board, we find LiDER to be capable of improving the performance of all base methods in all evaluated scenarios, both in terms of FAA and FF metrics. Most notably, we find it especially beneficial for methods that feature the most compelling results – usually DER++ and iCaRL –, suggesting that their higher generalization capability can still benefit from increased smoothness in latent space. Differently, scenarios where a method fail to prevent forgetting *i.e.* GDumb with a reduced buffer size usually feature a lower degree of benefit from LiDER but still a notable improvement. However, as it takes advantage both from increased buffers and pre-trained initialization, we observe a considerable performance gap with our proposal. Notably, on Split CIFAR-100, GDumb with a buffer size of 500 moves from a performance gain of $+0.98\%$ to $+6.46\%$ when increasing the buffer to 2000 samples and $+2.99\%$ with pre-train.

Finally, the average performance gain of LiDER (*i.e.*, $2.32\%$, $2.08\%$, and $4.36\%$ on Split CIFAR-100, Split *mini*ImageNet, and Split CUB-200 respectively) seems to confirm our intuitions.

Table 1: For different rehearsal approaches, Final Average Accuracy (FAA) [↑] on several benchmarks w/wo LiDER regularization.

| Benchmark | Split CIFAR-100 | | | | Split *mini*ImageNet | | Split CUB-200 | |
|---|---|---|---|---|---|---|---|---|
| **Pre-training** | ✗ | | *Tiny ImageNet* | | ✗ | | *ImageNet* | |
| Joint (UB) | 73.29 | | 75.20 | | 53.55 | | 78.54 | |
| Finetune | 9.29 | | 9.52 | | 4.51 | | 8.56 | |
| **Buffer Size** | 500 | 2000 | 500 | 2000 | 2000 | 5000 | 400 | 1000 |
| iCaRL [62] | 44.04 | 50.23 | 56.00 | 58.10 | 22.58 | 22.78 | 56.52 | 60.09 |
| + **LiDER** | 47.02 | 51.21 | 57.24 | 60.97 | 23.22 | 23.95 | 57.12 | 60.37 |
| DER++ [15] | 37.13 | 52.08 | 43.65 | 58.05 | 23.44 | 30.43 | 49.30 | 61.42 |
| + **LiDER** | 39.25 | 53.27 | 45.37 | 60.88 | 28.33 | 35.04 | 57.90 | 67.97 |
| X-DER - RPC [12] | 44.62 | 54.44 | 57.45 | 62.46 | 26.38 | 29.91 | 58.23 | 64.90 |
| + **LiDER** | 45.22 | 54.71 | 57.76 | 62.78 | 29.15 | 32.56 | 60.00 | 65.98 |
| GDumb [61] | 9.28 | 19.69 | 23.09 | 36.05 | 15.22 | 27.79 | 9.36 | 18.98 |
| + **LiDER** | 10.22 | 26.15 | 26.09 | 41.98 | 15.24 | 29.49 | 9.67 | 19.51 |
| ER-ACE [17] | 36.48 | 48.41 | 48.19 | 57.34 | 22.60 | 27.92 | 41.83 | 51.98 |
| + **LiDER** | 38.43 | 50.32 | 48.97 | 57.39 | 24.13 | 30.00 | 50.89 | 60.92 |

Table 2: Comparison between different regularization strategies (FAA, [↑]).

| Benchmark | Split CIFAR-100 | | Split miniImageNet | | Split CUB-200 | |
|---|---|---|---|---|---|---|
| **Pre-training** | ✗ | | ✗ | | *ImageNet* | |
| **Buffer Size** | 500 | 2000 | 2000 | 5000 | 400 | 1000 |
| ER-ACE | 36.48 | 48.41 | 22.60 | 27.92 | 41.83 | 51.98 |
| + sSGD | **39.59** | 49.70 | 22.43 | 24.12 | 22.67 | 29.88 |
| + oEwC | 35.06 | 45.59 | **24.32** | 29.46 | 48.34 | 59.74 |
| + oLAP | 36.58 | 47.66 | 23.19 | 28.77 | 42.64 | 52.86 |
| + **LiDER** | 38.43 | **50.32** | 24.13 | **30.00** | **50.89** | **60.92** |
| DER++ | 37.13 | 52.08 | 23.44 | 30.43 | 49.30 | 61.42 |
| + sSGD | 38.48 | 50.74 | 19.29 | 24.24 | 31.08 | 41.69 |
| + oEwC | 35.22 | 51.53 | 24.53 | 31.91 | 51.86 | 62.54 |
| + oLAP | 34.48 | 50.80 | 25.02 | 32.78 | 49.56 | 63.27 |
| + **LiDER** | **39.25** | **53.27** | **28.33** | **35.04** | **57.90** | **67.97** |

## 4.3 Comparison with regularization approaches

To provide a thorough evaluation, we investigate how the following existing regularization techniques (coupled with replay strategies such as ER-ACE and DER++) compare with our solution:

- **Stable SGD (sSGD)** [54], which introduces some specific alterations to the model's training regime that bias the optimization towards flat minima of the loss landscape;
- **online EwC (oEWC)** [65] and **Online Laplace (oLAP)** [64], applying Hessian-based parameter-importance estimation to constrain the most significant model parameters for previous tasks;

As reported in Tab. 2, sSGD boosts ER-ACE and DER++ only on Split CIFAR-100; whereas, its performance degrades severely both on the more complex Split *mini*ImageNet and on Split CUB-200, where we suspect it fails to effectively exploit the pre-trained network. By contrast, we find the application of oEwC and oLAP to be rewarding, especially in the presence of pre-training. In this respect, we recall that pre-training has a known flattening effect on the loss landscape [53]; in such a setting, encouraging the model to stay close to its prior (as done by these methods) could be the key to explain the improvements they carry out. Concerning our approach, it proves almost always more effective than any of the other tested approaches.

Table 3: FAA after poisoning the buffer for DER++ with and without LiDER.

| $p$ | DER++ | +LiDER |
|------|-------|--------|
| .0% | 37.14 | **39.25** |
| .01% | 36.13 | **38.08** |
| .1% | 31.35 | **35.53** |
| .25% | 28.74 | **30.78** |

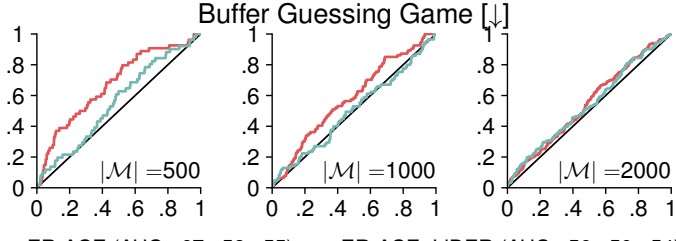

Figure 2: ROC curves for the Buffer Guessing Game, showing the likelihood of a given sample belonging in $\mathcal{M}$.

## 5  Model analysis

### 5.1  On the memory buffer

**Robustness to buffer poisoning**. To evaluate the performance of a ML model, practitioners usually resort to synthetic benchmarks that simulate the progressive arrival of novel knowledge. While this methodology is useful to compare the effectiveness of new proposals a key element is usually neglected: the labels given to each sample may – and in the real-world usually does – be incorrect. Notably, this aspect is even more relevant in a CL scenario and especially when a rehearsal strategy is employed. As previously shown, samples included in the buffer are the ones that are most likely to be overfitted, which would induce a severe loss of performance if the label is incorrect.

In the following, we assess the harmful effect of mislabeled examples included in the memory buffer: namely, we randomly assign with probability $p$ a new label to the samples as these are added in the buffer (*i.e.*, *label poisoning*). To better suit a CL scenario, poisoned labels are chosen from those of the current task. Results reported in Tab. 3 show the evaluation performed on top of DER++ for the Split CIFAR-100 benchmark. As one could expect, performance degrades as $p$ increases; however, we also observe that LiDER retains a higher level of accuracy against poisoning. This effect suggests that our proposal leads to a looser decision boundary around the items stored in the buffer.

**Buffer Guessing game**. We propose a novel experiment to further illustrate the tendency of rehearsal-based CL models to overfitting. As posited in Sec. 1, a substantially different regime affects replay and stream examples: the former plays a much larger role in shaping the decision boundary w.r.t. the latter. To validate this intuition, we propose a simple *buffer guessing game*: given a rehearsal-based model $f$ fully trained on a CL benchmark and the dataset $\tau_0$ used in its first encountered task, we aim to find $\mathcal{M} \cap \tau_0$ (*i.e.* the subset of data-points that are included in the model's buffer).

We approach the game by associating each $x \in \tau_0$ with a score $s_x$ computed in a neighborhood of $x$. Such a score quantifies the mean *height* of the decision surface, *i.e.*, the difference between the probability of the right class and the highest one. As specified in [87], we model the neighborhood by leveraging random perturbations; moreover, we compute $s_x$ w.r.t. to the TIL prediction function in order to avoid the influence of inter-task biases on our results. Finally, to assess whether these scores can help separating in-buffer examples from all the other, we evaluate the ROC curve obtained from these scores. Fig. 2 reports the results for ER-ACE and ER-ACE+LiDER on Split CIFAR-100 across different buffer sizes. We see that: *i)* ER-ACE makes it easier to reconstruct the content of the buffer, as indicated by larger ROC-AUC scores w.r.t. ER-ACE+LiDER; *ii)* in line with our expectations, this effect is increased when employing smaller memory buffers, as this leads to the repeated optimization of a smaller pool of data.

### 5.2  Applying LiDER on current task's examples

Our approach devotes a regularization specific for buffer datapoints; however, it could be investigated if its benefits extend also to the current task. In this respect, we have performed several experiments switching the target of regularization: not the examples from the replay memory, but those belonging to the current task. Tab. 4 reports the results: while the original approach delivers consistent improvements, applying LiDER on incoming examples delivers inferior results.

Table 4: Comparison (FAA) between two possible targets of regularization: examples from the current task (stream) *vs* buffer datapoints (standard LiDER).

| Benchmark | Split CIFAR-100 | | | | Split *mini*ImageNet | | Split CUB-200 | |
|---|---|---|---|---|---|---|---|---|
| Pre-training | ✗ | | *Tiny ImageNet* | | ✗ | | *ImageNet* | |
| Buffer Size | 500 | 2000 | 500 | 2000 | 2000 | 5000 | 400 | 1000 |
| ER-ACE | 36.48 | 48.41 | 48.19 | 57.34 | 22.60 | 27.92 | 41.83 | 51.98 |
| + LiDER (curr. task) | 37.54 | **50.37** | 48.94 | 57.07 | 23.35 | 29.25 | 48.44 | 59.60 |
| + LiDER (buffer) | **38.43** | 50.32 | **48.97** | **57.39** | **24.13** | **30.00** | **50.89** | **60.92** |
| DER++ | 37.13 | 52.08 | 43.65 | 58.05 | 23.44 | 30.43 | 49.30 | 61.42 |
| + LiDER (curr. task) | 34.78 | 49.76 | 44.48 | 59.39 | 24.84 | 31.05 | 56.96 | 66.63 |
| + LiDER (buffer) | **39.25** | **53.27** | **45.37** | **60.88** | **28.33** | **35.04** | **57.90** | **67.97** |

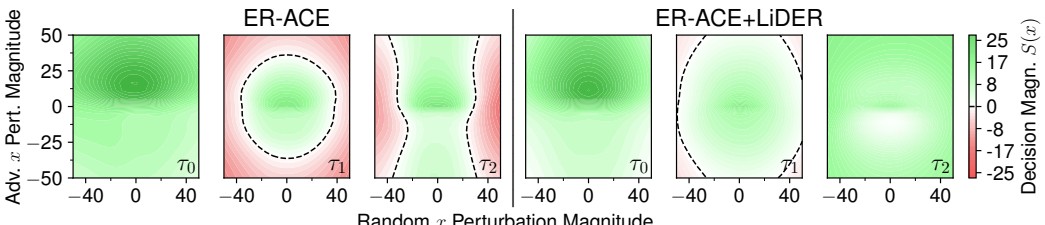

Figure 3: Effect of our proposal on the robustness of the decision boundary produced by ER-ACE across subsequent tasks. Higher values (in green) indicate areas of high confidence and span the possible directions where perturbations are not disruptive (best seen in color).

Such an evidence can be explained by reviewing the distinct data regimes the current and previous tasks are subject to. While learning the current task, indeed, the model can access many and many samples from its underlying distribution; therefore, the epistemic uncertainty [38] reduces and the learned decision boundaries are likely to be smoother. In this case, the Lipschitz regularization could represent an overkill, threatening to restrain the learning with no advantages. In our formulation, instead, only few examples are available for retaining the knowledge of past tasks: the risk of progressive overfitting – which we expressed through the progressive degradation of decision boundaries – is more severe: therefore, tailored countermeasures are more likely to be effective.

### 5.3 Generalization measures

**Decision surface of LiDER**. Fig. 1 presents depicts the model's tolerance to input perturbations in the form of a decision surface plot [87]. Such a visualization is constructed by focusing on a set of perturbations $x_p \triangleq x + i \cdot \alpha + j \cdot \beta$ computed around a data-point $x$ ($\alpha$ is a random divergence direction and $\beta$ corresponds to the direction induced by the first step of a non-targeted FGSM attack [44]). The plot shows the respective values of the decision function $S(x_p)$, where $S(x) \triangleq f(x)_t - \max_{i \neq t} f(x)_i$ (with $f(\cdot)$ indicating pre-softmax responses) and highlights decision boundary of the model (*i.e.* the locus of $\{x_p | S(x_p) = 0\}$), in correspondence of which the model accuracy fails.

In Fig. 3, we adopt the same approach to compare the decision boundaries around rehearsed $\tau_0$ examples for ER-ACE with and without LiDER. While both models start with a similar robust decision landscape in $\tau_0$, later tasks reveal a clear shrinking behavior in ER-ACE. On the contrary, introducing $L$-based regularization lead to minimal decision boundary deterioration in later tasks.

**Loss Landscape of LiDER**. Previous CL works proposed investigating the generalization capabilities of a CL model by evaluating the flatness of its attained minima [15, 12]. We likewise evaluate the effect of LiDER both in terms of its resilience to weight perturbations [57, 39] and the eigenvalues of the Hessian of the loss function [39, 34] in Fig. 4. We see that DER++ and ER-ACE combined with LiDER see an improvement of both metrics. This is expected, as the Lipschitz constant of a DNN has been commonly interpreted as a generalization measure [82, 72].

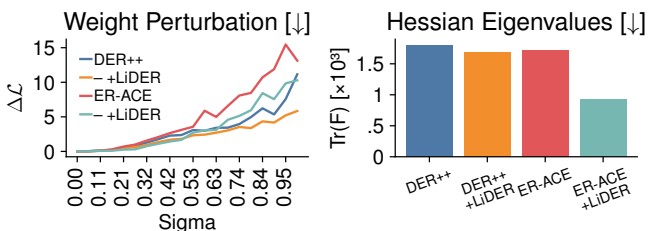

**Weight Perturbation [↓]**    **Hessian Eigenvalues [↓]**

Table 5: Ablation: fixed Lipschitz targets *vs* learned targets.

| | Model | $|\mathcal{M}|$ | Fixed tgt | *Eq. 9* |
|---|---|---|---|---|
| *w/o pre-tr.* | DER++ | 500 | 36.42 | **39.25** |
| | ER-ACE | 500 | 34.99 | **38.43** |
| | DER++ | 2000 | 51.52 | **53.27** |
| | ER-ACE | 2000 | 46.70 | **48.97** |
| *pre-tr.* | DER++ | 500 | 43.16 | **45.37** |
| | ER-ACE | 500 | 45.21 | **48.97** |
| | DER++ | 2000 | 59.53 | **60.68** |
| | ER-ACE | 2000 | 54.82 | **57.39** |

Figure 4: (Left) Robustness of models regularized with LiDER against weight perturbations (best seen in color). (Right) Flatness around the minima found during optimization, measured as the sum of the eigenvalues of the Hessian matrix.

### 5.4 Optimization with a fixed target

Instead of constraining the optimization of the Lipschitz targets of Eq. 7 by an additional contribution, one might argue that a similar effect could be obtained by fixing an initial value and force the model to reduce its capacity until it meets the desiderata [48]. In Tab. 5 we empirically show that such approach does not lead to satisfactory results in a CL setting, with our proposal consistently exceeding its performance on various settings and base methods.

We finally refer to the supplementary materials for additional studies *e.g.*, the evaluation of the single-epoch scenario [49], the efficiency-accuracy trade-off, and a sensitivity analysis of hyperparameters.

## 6 Conclusions

We present LiDER, a novel regularization strategy to compensate for the phenomenon of buffer overfitting in rehearsal-based Continual Learning; it enforces the decision boundary of replayed samples to be smoother by bounding the complexity of the model. We show that such approach can be readily applied to diverse state-of-the-art models and remains competitive across various settings. Finally, we provide an extensive investigation to assess the validity of theoretical assumptions.

## Limitations & Societal Impact

Our approach relies on a coarse estimation of the Lipschitz constant and, in particular, on its upper bound (the exact computation is a NP-hard problem). We highlight the tighter bounds have been proposed [35, 33], at the expense of higher complexity, which could clash with the incremental scenario subject of our work. Moreover, our approximation cannot be applied to not-Lipschitz continuous layers (*e.g.*, cross-attention) [41]. In the following, we briefly recap the notions through which our approximation can be favorably refined, leaving for future works the investigation of the efficiency/accuracy trade-off inherent in these advanced estimations.

It could help to provide, for each activation function $\sigma^l$, tighter upper bounds of its Lipschitz constant $||\sigma^l||_L$. It is noted that our assumption $||\sigma^l||_L \leq 1$ refers to the global 1-Lipschitz continuity, which requires the inequality to hold for all points from the input domain. Such a requirement is too strict and not representative of DNNs, as the input of a layer does not distribute uniformly but is often constrained in a subspace, whose shape depends on the activations of the previous layers. For this reason, recent works [35, 33] refer to a different property *i.e.*, the local Lipschitz continuity, which bounds output perturbation only for certain input regions. Notably, it has been proved to provide a tighter upper bound (see Theorem 1 of [33]), and hence a more effective regularizing signal.

Concerning societal impact, we do not feel that this work will have detrimental applications that might affect any public. We only remark that, as our approach is based on rehearsal techniques that store in-plain data, it cannot be used in those scenarios where privacy constraints are crucial.

**Acknowledgement**. This paper has been supported by: Italian Ministerial grant PRIN 2020 "LEGO.AI: LEarning the Geometry of knOwledge in AI systems", n. 2020TA3K9N; FF4EuroHPC: HPC Innovation for European SMEs, Project Call 1. FF4EuroHPC has received funding from the European High-Performance Computing Joint Undertaking (JU) under grant agreement n. 951745.

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
