# On the Effectiveness of Lipschitz-Driven Rehearsal in Continual Learning – Supplementary Material

Lorenzo Bonicelli[1]    Matteo Boschini[1]    Angelo Porrello[1]
Concetto Spampinato[2]    Simone Calderara[1]

[1]AImageLab - University of Modena and Reggio Emilia
[2]PeRCeiVe Lab - University of Catania

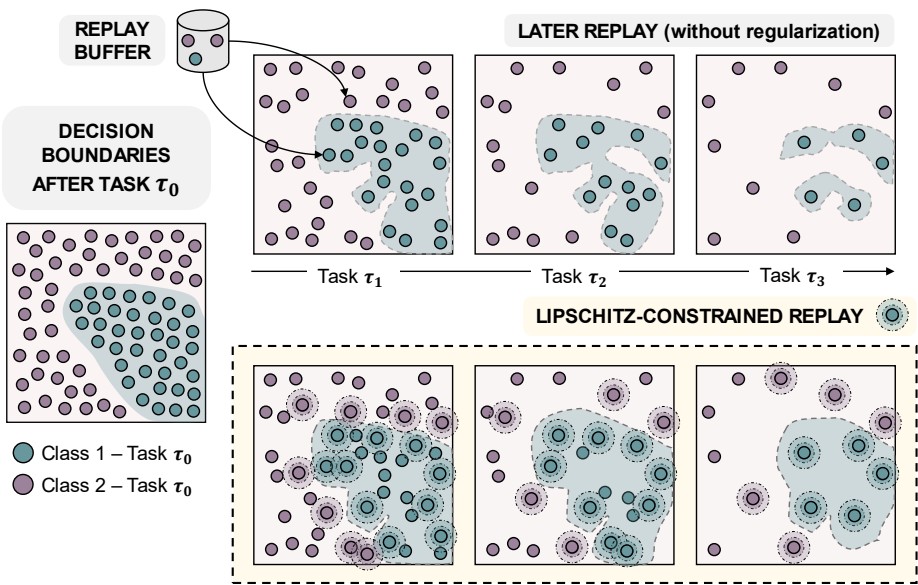

Figure 1: **Illustration of the Effect of LiDER**. Left: depiction of an initial decision boundary learned by the model after task $\tau_0$. Right (first row): in subsequent tasks $\tau_1 \rightarrow \tau_3$, classical rehearsal approaches can access a decreasing amount of examples from their replay buffer: hence, overfitting shows as erosion of the initial boundaries. When applying Lipschitz-based constraints on replayed data (second row), small output variations are required around replay data, thus favoring less curved boundaries.

## 1 Experiments

### 1.1 Additional details on the experimental settings

Our selection of benchmarks involves assessing the performance of the included models on a variety of scenarios, with a part of them involving starting from a pre-trained backbone. For the pre-trained **Split CIFAR-100** benchmark, we initially train a ResNet18 backbone on images from *Tiny ImageNet*, resized to $32 \times 32$ for compatibility with the ones from CIFAR-100. We opt for SGD as optimizer and train for 50 epochs, reducing the learning rate by a factor of 2 at epochs 20, 30, 40, and 45 starting from an initial value of 0.1. The final linear classifier is later discarded and reinitialized with the appropriate number of classes. During CL training, either with or without pre-train, we train on each

Table 1: For different rehearsal approaches, Final Forgetting (FF) [↓] on several benchmarks w/wo LiDER regularization.

| Benchmark | Split CIFAR-100 | | | | Split *mini*ImageNet | | Split CUB-200 | |
|---|---|---|---|---|---|---|---|---|
| **Pre-training** | ✗ | | *Tiny ImageNet* | | ✗ | | *ImageNet* | |
| Finetune | 86.62 | | 92.31 | | 77.38 | | 82.38 | |
| **Buffer Size** | 500 | 2000 | 500 | 2000 | 2000 | 5000 | 400 | 1000 |
| iCaRL [10] | 21.70 | 17.92 | 19.27 | 16.89 | 16.46 | 16.37 | 13.43 | 11.41 |
| + **LiDER** | 21.89 | 17.13 | 19.16 | 15.49 | 11.21 | 11.18 | 14.31 | 10.89 |
| DER++ [3] | 49.80 | 31.10 | 48.72 | 29.65 | 46.69 | 37.11 | 36.05 | 19.95 |
| + **LiDER** | 45.50 | 27.51 | 48.16 | 25.16 | 36.29 | 25.02 | 27.55 | 14.44 |
| X-DER - RPC [2] | 31.84 | 17.01 | 16.86 | 12.07 | 38.33 | 28.29 | 16.58 | 9.03 |
| + **LiDER** | 28.38 | 11.33 | 11.33 | 11.26 | 27.18 | 20.59 | 15.64 | 8.64 |
| ER-ACE [4] | 38.21 | 27.90 | 31.84 | 25.48 | 23.74 | 19.72 | 26.42 | 18.79 |
| + **LiDER** | 36.00 | 28.30 | 28.58 | 25.37 | 25.97 | 19.99 | 20.79 | 14.62 |

Table 2: Comparison between different regularization strategies (FF, [↓]).

| Benchmark | Split CIFAR-100 | | Split miniImageNet | | Split CUB-200 | |
|---|---|---|---|---|---|---|
| **Pre-training** | ✗ | | ✗ | | *ImageNet* | |
| **Buffer Size** | 500 | 2000 | 2000 | 5000 | 400 | 1000 |
| ER-ACE | 38.21 | 27.90 | 23.74 | 19.72 | 26.42 | 18.79 |
| + sSGD | **39.59** | 24.44 | 13.99 | 11.02 | 21.18 | 14.24 |
| + oEwC | 38.08 | 27.55 | **24.32** | 20.11 | 27.59 | 17.36 |
| + oLAP | 37.88 | 29.34 | 28.69 | 21.85 | 29.57 | 19.24 |
| + **LiDER** | 36.00 | **50.32** | 25.97 | **30.00** | **50.89** | **60.92** |
| DER++ | 49.80 | 31.10 | 46.69 | 37.11 | 36.05 | 19.95 |
| + sSGD | 39.70 | 25.56 | 18.73 | 28.16 | 30.44 | 21.96 |
| + oEwC | 52.13 | 32.18 | 47.90 | 36.35 | 30.14 | 16.90 |
| + oLAP | 55.84 | 35.14 | 40.65 | 32.92 | 32.68 | 15.90 |
| + **LiDER** | **39.25** | **53.27** | **28.33** | **35.04** | **57.90** | **67.97** |

task with SGD for 50 epochs and decay the learning rate by a factor of 10 at epochs 35 and 45; we keep the same size for the batch drawn from the stream and from the buffer at 64 items.

For experiments involving **Split CUB-200**, pre-train on ImageNet is carried by employing the initialization provided by `torchvision`[1] for the ResNet50 backbone. We then follow by training on the tasks for 50 epochs each, with 16 samples as the size of the batch both for stream and buffer.

Finally, experiments on *mini***ImageNet** do no feature pre-training; we train each task for 80 epochs and decay the learning rate with a factor of 0.2 at epochs 35, 60, and 75. We keep a consistent batch size of 128 items for stream and buffer.

## 1.2 Final Forgetting (FF)

For all the experiments reported in the main document, we hereinafter report the Final Forgetting (FF) [5] metric (see Tab. 1, Tab. 2, and Tab. 3), formally defined as:

$$\text{FF} = \frac{1}{|T| - 1} \sum_{i=0}^{|T|-2} \max_{t \in \{0, \dots, |T|-2\}} \{a_i^t - a_i^{|T|-1}\}, \tag{1}$$

where $a_i^t$ indicates the accuracy on task $\tau_i$ after training on the $t^{th}$ task.

---
[1] https://pytorch.org/vision/stable/index.html

Table 3: Comparison between two possible targets of regularization (FF, [↓]).

| Benchmark | Split CIFAR-100 | | | | Split *mini*ImageNet | | Split CUB-200 | |
|---|---|---|---|---|---|---|---|---|
| Pre-training | ✗ | | *Tiny ImageNet* | | ✗ | | *ImageNet* | |
| Buffer Size | 500 | 2000 | 500 | 2000 | 2000 | 5000 | 400 | 1000 |
| ER-ACE | 38.21 | 27.90 | 31.84 | 25.48 | 23.74 | 19.72 | 26.42 | 18.79 |
| + LiDER **(curr. task)** | 36.46 | 27.24 | 28.80 | 23.50 | 23.56 | 18.78 | 25.16 | 15.21 |
| + LiDER **(buffer)** | 36.00 | 28.30 | 28.58 | 25.37 | 25.97 | 19.99 | 20.79 | 14.62 |
| DER++ | 49.80 | 31.10 | 48.72 | 29.65 | 46.69 | 37.11 | 36.05 | 19.95 |
| + LiDER **(curr. task)** | 54.41 | 35.91 | 49.11 | 28.94 | 44.54 | 34.85 | 28.42 | 16.30 |
| + LiDER **(buffer)** | 45.50 | 27.51 | 48.16 | 25.16 | 36.29 | 25.02 | 27.55 | 14.44 |

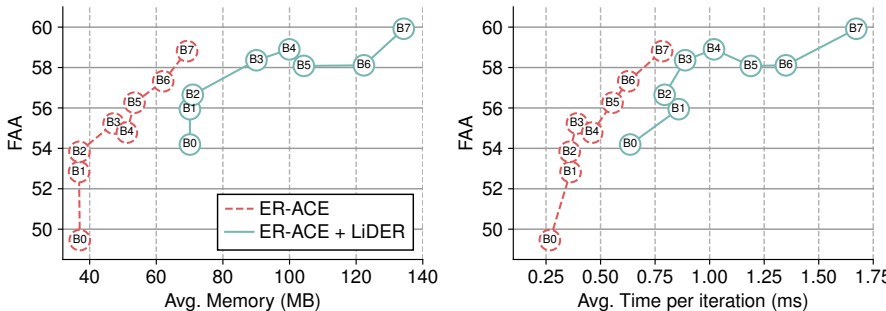

Figure 2: *Memory footprint/accuracy* (left) and *speed/accuracy* (right) trade-offs of our approach in combination with ER-ACE. Results are reported for increasingly complex backbone networks (ranging from EffecientNET-B0 to -B7).

## 1.3 On the efficiency *vs* accuracy trade-offs

We have benchmarked the *memory footprint/accuracy* and *speed/accuracy* trade-offs of our approach in combination with ER-ACE, for increasing complex backbone networks (ranging from EffecientNET-B0 to the deepest EffecientNET-B7). The experimental evaluation – whose results are shown in Fig. 2 – has been carried out on Split CUB-200, the dataset with the highest input resolution in our tests. For a meaningful term of comparison, we also report the performance trend of ER-ACE without LiDER.

As can be observed, our approach clearly involves an overhead, but this is fully rewarded by superior accuracy, especially for smaller architectures (EfficientNet-B≤4). As a final note, while our use of power iteration might seem a major hindrance to scalability, we observed in practice that few iterations were enough to obtain good and stable estimates of the eigenvalues.

## 1.4 Single-epoch setting

Several Continual Learning works focus on the **online** scenario, which allows the model to observe each task only for one epoch [1, 7, 9, 6, 8]. The investigation of the online setting is certainly worth-noting; however, we advocate for what has been said in [3]: when only one epoch is allowed on the current task, even the pure-SGD baseline fails at fitting it with adequate accuracy, especially with complex datasets such as CIFAR-100 and *mini*ImageNet. Therefore, the resulting performance – and in turn the comparisons among different approaches – can be difficult to read, as the effects of catastrophic forgetting and those linked to underfitting interleave here. Furthermore, it is even more complicated comparing approaches that were conceived only in either of the two settings (multi-epochs *vs.* single-epoch) such as ours and GMED [8].

We therefore exhort the reader to interpret cautiously the results provided in Tab. 4, reporting the single-epoch performance of various methods equipped with our regularizer. At a first glance,

Table 4: Single-epoch evaluation setting. Results reported as FAA (FF).

| Benchmark | Split CIFAR-100 - 20 Tasks | | Split CIFAR-100 - 10 Tasks | |
|---|---|---|---|---|
| Pre-training | ✗ | | ✗ | |
| Buffer Size | 500 | 2000 | 500 | 2000 |
| GMED | **22.39** (66.89) | **36.18** (49.97) | **23.69** (58.22) | **34.07** (45.47) |
| DER++ | 14.03 (51.61) | 19.33 (42.43) | 10.49 (41.46) | 23.56 (26.73) |
| + LiDER | 15.54 (50.34) | 21.93 (44.78) | 14.65 (43.58) | 24.36 (28.50) |
| ER-ACE | 17.58 (11.79) | 21.60 (10.17) | 17.23 (10.24) | 21.73 (05.69) |
| + LiDER | 18.28 (11.66) | 25.19 (09.78) | 18.98 (09.56) | 24.86 (03.83) |

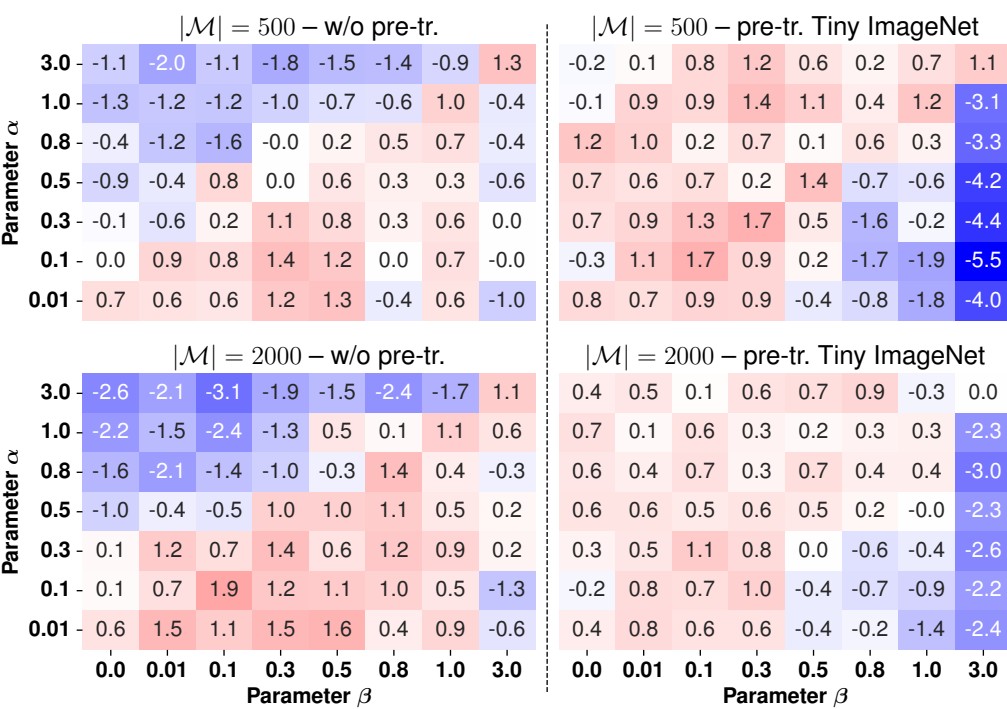

Figure 3: Sensitivity analysis of ER-ACE + LiDER to the hyperparameters $\alpha$ and $\beta$ on Split CIFAR-100. Results for different sizes of the memory buffer, with and without pre-training.

the results provided herein show a remarkable improvement for both DER++ and ER-ACE when equipped with LiDER, with a respective average gain of 2.27% and 2.29%. While consistent, however, the performance gain is not sufficient to make the baseline methods competitive against a method specifically designed for the online setting, such as GMED.

## 2 Hyperparameters

### 2.1 Sensitivity analysis

Fig. 3 proposes a 2D summary of the performance variation yielded by different values of $\alpha$ and $\beta$ (introduced by Eq. 9 of the main paper). In particular, each item of these matrices reports, for a given combination, the difference w.r.t. the average measured FAA. As can be observed, we distinguish

two separate regimes: one for the cases when models are trained from scratch; another when using a pre-trained model.

Without pre-training, we obtain a performance gain if the two parameters are comparable, with $\beta \geq \alpha$. If $\alpha > \beta$, we are overemphasizing the contribution of the first term of Eq. 9 (which brings each layer's $\lambda_1^k$ and $c_k$ close to each other) over the second one (which induces small Lipschitz targets). For a randomly initialized model, this may lead the initial value of $\lambda_1^k$ to mislead $c_k$. Differently, if $\beta \gg \alpha$, the model is encouraged to over-regularize its response, resulting in a reduced capability of fitting the encountered data.

Instead, if the backbone is pre-trained, we see an overall much stabler behavior, due to the $\lambda_1^k$ for each layer being well-behaved from the beginning of training. Here, the only pitfall is given by $\beta \gg \alpha$, which again leads the model to oversmoothing.

## 2.2 Hyperparameter Search

### 2.2.1 CIFAR-100 w/o pre-tr. – best values

**Joint**: $lr$:0.3
**Finetune**: $lr$:0.01
**Buffer 500**
**iCaRL**: $lr$:0.1, wd:1e-05
**iCaRL + LiDER**: $lr$:1.0, wd:1e-05, $\alpha_{LiDER}$:0.01, $\beta_{LiDER}$:0.01
**DER++**: $lr$:0.1, $\alpha_{\text{DER++}}$:0.1, $\beta_{\text{DER++}}$:0.5
**DER++ + LiDER**: $lr$:0.1, $\alpha_{\text{DER++}}$:0.3, $\beta_{\text{DER++}}$:0.3, $\alpha_{LiDER}$:0.3, $\beta_{LiDER}$:0.1
**GDumb**: wd:1e-06, Epochs$_{\text{Fitting}}$:250.0, $\alpha_{\text{Cutmix}}$:1.0, lr$_{\text{min}}$:0.0005, lr$_{\text{max}}$:0.05
**GDumb + LiDER**: wd:1e-06, Epochs$_{\text{Fitting}}$:250.0, $\alpha_{\text{Cutmix}}$:1.0, lr$_{\text{min}}$:0.0005, lr$_{\text{max}}$:0.05, $\alpha_{LiDER}$:0.01, $\beta_{LiDER}$:0.01
**ER-ACE**: $lr$:0.03
**ER-ACE + LiDER**: $lr$:0.1, $\alpha_{LiDER}$:0.1, $\beta_{LiDER}$:0.3
**Buffer 2000**
**iCaRL**: $lr$:0.03, wd:1e-05
**iCaRL + LiDER**: $lr$:1.0, wd:1e-05, $\alpha_{LiDER}$:0.01, $\beta_{LiDER}$:0.001
**DER++**: $lr$:0.03, $\alpha_{\text{DER++}}$:0.3, $\beta_{\text{DER++}}$:0.3
**DER++ + LiDER**: $lr$:0.1, $\alpha_{\text{DER++}}$:0.2, $\beta_{\text{DER++}}$:0.5, $\alpha_{LiDER}$:0.01, $\beta_{LiDER}$:0.1
**GDumb**: wd:5e-05, Epochs$_{\text{Fitting}}$:250.0, $\alpha_{\text{Cutmix}}$:1.0, lr$_{\text{min}}$:0.0005, lr$_{\text{max}}$:0.05
**GDumb + LiDER**: wd:1e-06, Epochs$_{\text{Fitting}}$:250.0, $\alpha_{\text{Cutmix}}$:1.0, lr$_{\text{min}}$:0.0005, lr$_{\text{max}}$:0.05, $\alpha_{LiDER}$:0.3, $\beta_{LiDER}$:0.01
**ER-ACE**: $lr$:0.03
**ER-ACE + LiDER**: $lr$:0.1, $\alpha_{LiDER}$:0.5, $\beta_{LiDER}$:0.01

### 2.2.2 CIFAR-100 w/ pre-tr. – best values

**Joint**: $lr$:3.0
**Finetune**: $lr$:0.1
**Buffer 500**
**iCaRL**: $lr$:1.0, wd:1e-05
**iCaRL + LiDER**: $lr$:1.0, wd:1e-05, $\alpha_{LiDER}$:0.01, $\beta_{LiDER}$:0.01
**DER++**: $lr$:0.1, $\alpha_{\text{DER++}}$:0.3, $\beta_{\text{DER++}}$:1.2
**DER++ + LiDER**: $lr$:0.1, $\alpha_{\text{DER++}}$:0.2, $\beta_{\text{DER++}}$:0.3, $\alpha_{LiDER}$:0.1, $\beta_{LiDER}$:0.1
**GDumb**: wd:1e-6, Epochs$_{\text{Fitting}}$:250.0, $\alpha_{\text{Cutmix}}$:1.0, lr$_{\text{min}}$:0.0005, lr$_{\text{max}}$:0.05
**GDumb + LiDER**: wd:1e-06, Epochs$_{\text{Fitting}}$:250.0, $\alpha_{\text{Cutmix}}$:1.0, lr$_{\text{min}}$:0.0005, lr$_{\text{max}}$:0.05, $\alpha_{LiDER}$:0.1, $\beta_{LiDER}$:0.01
**ER-ACE**: $lr$:0.03
**ER-ACE + LiDER**: $lr$:0.03, $\alpha_{LiDER}$:0.1, $\beta_{LiDER}$:0.3
**Buffer 2000**
**iCaRL**: $lr$:1.0, wd:1e-05
**iCaRL + LiDER**: $lr$:1.0, wd:1e-05, $\alpha_{LiDER}$:0.01, $\beta_{LiDER}$:0.1
**DER++**: $lr$:0.1, $\alpha_{\text{DER++}}$:0.5, $\beta_{\text{DER++}}$:0.1

**DER++ + LiDER**: $lr$:0.1, $\alpha_{\text{DER++}}$:0.3, $\beta_{\text{DER++}}$:0.3, $\alpha_{LiDER}$:0.1, $\beta_{LiDER}$:0.1
**GDumb**: wd:1e-06, $\text{Epochs}_{\text{Fitting}}$:250.0, $\alpha_{\text{Cutmix}}$:1.0, $\text{lr}_{\text{min}}$:0.0005, $\text{lr}_{\text{max}}$:0.05
**GDumb + LiDER**: wd:1e-06, $\text{Epochs}_{\text{Fitting}}$:250.0, $\alpha_{\text{Cutmix}}$:1.0, $\text{lr}_{\text{min}}$:0.0005, $\text{lr}_{\text{max}}$:0.05, $\alpha_{LiDER}$:0.1, $\beta_{LiDER}$:0.01
**ER-ACE**: $lr$:0.1
**ER-ACE + LiDER**: $lr$:0.1, $\alpha_{LiDER}$:0.3, $\beta_{LiDER}$:0.3

### 2.2.3 *mini*ImageNet – best values

**Joint**: $lr$:0.1
**Finetune**: $lr$:0.03
**Buffer 2000**
**iCaRL**: $lr$:0.1, wd:0.0
**iCaRL + LiDER**: $lr$:0.3, wd:1e-05, $\alpha_{LiDER}$:0.01, $\beta_{LiDER}$:0.01
**DER++**: $lr$:0.1, $\alpha_{\text{DER++}}$:0.3, $\beta_{\text{DER++}}$:0.8
**DER++ + LiDER**: $lr$:0.1, $\alpha_{\text{DER++}}$:0.3, $\beta_{\text{DER++}}$:0.3, $\alpha_{LiDER}$:0.1, $\beta_{LiDER}$:0.1
**GDumb**: wd:5e-05, $\text{Epochs}_{\text{Fitting}}$:250.0, $\alpha_{\text{Cutmix}}$:1.0, $\text{lr}_{\text{min}}$:0.0005, $\text{lr}_{\text{max}}$:0.05
**GDumb + LiDER**: wd:0.0, $\text{Epochs}_{\text{Fitting}}$:250.0, $\alpha_{\text{Cutmix}}$:1.0, $\text{lr}_{\text{min}}$:0.0005, $\text{lr}_{\text{max}}$:0.05, $\alpha_{LiDER}$:0.01, $\beta_{LiDER}$:0.3
**ER-ACE**: $lr$:0.1
**ER-ACE + LiDER**: $lr$:0.1, $\alpha_{LiDER}$:0.3, $\beta_{LiDER}$:0.01
**Buffer 5000**
**iCaRL**: $lr$:0.1, wd:0.0
**iCaRL + LiDER**: $lr$:0.1, wd:0.0, $\alpha_{LiDER}$:0.1, $\beta_{LiDER}$:0.01
**DER++**: $lr$:0.1, $\alpha_{\text{DER++}}$:0.3, $\beta_{\text{DER++}}$:0.8
**DER++ + LiDER**: $lr$:0.1, $\alpha_{\text{DER++}}$:0.3, $\beta_{\text{DER++}}$:0.3, $\alpha_{LiDER}$:0.1, $\beta_{LiDER}$:0.3
**GDumb**: wd:5e-05, $\text{Epochs}_{\text{Fitting}}$:250.0, $\alpha_{\text{Cutmix}}$:1.0, $\text{lr}_{\text{min}}$:0.0005, $\text{lr}_{\text{max}}$:0.05
**GDumb + LiDER**: wd:0.0, $\text{Epochs}_{\text{Fitting}}$:250.0, $\alpha_{\text{Cutmix}}$:1.0, $\text{lr}_{\text{min}}$:0.0005, $\text{lr}_{\text{max}}$:0.05, $\alpha_{LiDER}$:0.01, $\beta_{LiDER}$:0.01
**ER-ACE**: $lr$:0.1
**ER-ACE + LiDER**: $lr$:0.1, $\alpha_{LiDER}$:0.3, $\beta_{LiDER}$:0.3

### 2.2.4 CUB-200 – best values

**Joint**: $lr$:0.1
**Finetune**: $lr$:0.1
**Buffer 400**
**iCaRL**: $lr$:0.1, wd:1e-05
**iCaRL + LiDER**: $lr$:0.3, wd:0.0, $\alpha_{LiDER}$:0.001, $\beta_{LiDER}$:0.001
**DER++**: $lr$:0.1, $\alpha_{\text{DER++}}$:0.5, $\beta_{\text{DER++}}$:0.5
**DER++ + LiDER**: $lr$:0.03, $\alpha_{\text{DER++}}$:0.5, $\beta_{\text{DER++}}$:0.8, $\alpha_{LiDER}$:0.1, $\beta_{LiDER}$:0.03
**GDumb**: wd:0.0, $\text{Epochs}_{\text{Fitting}}$:250.0, $\alpha_{\text{Cutmix}}$:1.0, $\text{lr}_{\text{min}}$:0.0005, $\text{lr}_{\text{max}}$:0.05
**GDumb + LiDER**: wd:1e-06, $\text{Epochs}_{\text{Fitting}}$:250.0, $\alpha_{\text{Cutmix}}$:1.0, $\text{lr}_{\text{min}}$:0.0005, $\text{lr}_{\text{max}}$:0.05, $\alpha_{LiDER}$:0.01, $\beta_{LiDER}$:0.5
**ER-ACE**: $lr$:0.1
**ER-ACE + LiDER**: $lr$:0.01, $\alpha_{LiDER}$:0.01, $\beta_{LiDER}$:0.1
**Buffer 1000**
**iCaRL**: $lr$:0.1, wd:1e-05
**iCaRL + LiDER**: $lr$:0.3, wd:0.0, $\alpha_{LiDER}$:0.001, $\beta_{LiDER}$:0.01
**DER++**: $lr$:0.1, $\alpha_{\text{DER++}}$:0.5, $\beta_{\text{DER++}}$:0.5
**DER++ + LiDER**: $lr$:0.03, $\alpha_{\text{DER++}}$:0.5, $\beta_{\text{DER++}}$:0.8, $\alpha_{LiDER}$:0.1, $\beta_{LiDER}$:0.1
**GDumb**: wd:5e-05, $\text{Epochs}_{\text{Fitting}}$:250.0, $\alpha_{\text{Cutmix}}$:1.0, $\text{lr}_{\text{min}}$:0.005, $\text{lr}_{\text{max}}$:0.05
**GDumb + LiDER**: wd:1e-06, $\text{Epochs}_{\text{Fitting}}$:250.0, $\alpha_{\text{Cutmix}}$:1.0, $\text{lr}_{\text{min}}$:0.0005, $\text{lr}_{\text{max}}$:0.05, $\alpha_{LiDER}$:0.3, $\beta_{LiDER}$:0.01
**ER-ACE**: $lr$:0.1
**ER-ACE + LiDER**: $lr$:0.01, $\alpha_{LiDER}$:0.3, $\beta_{LiDER}$:0.3