# OpenReview forum: "On the Effectiveness of Lipschitz-Driven Rehearsal in Continual Learning"
_NeurIPS.cc/2022/Conference — NeurIPS 2022 Accept_

### Official Review · Reviewer_PLZ4 · 2022-07-02

**Rating:** 5
**Confidence:** 5
**Soundness:** 3 good
**Presentation:** 3 good
**Contribution:** 2 fair

**Summary:**

In this work authors propose LIDER - an extension to the existing continual learning methods with buffer replay, through additional regularisation based on Lipschitz constant that reduces overfitting to examples stored in the buffer, by improving the generalisation of the classifier. In experimental studies LIDER improves the performance of recent state-of-the-art methods.

**Questions:**

- In this submission a second - additional regularisation term is introduced that forces Lipschitz constants of individual layers to be as close to 0 as possible. As stated in the work, employment of this regularisation should lead to lower sensitivity to changes in the input. I miss the direct link, why this should prevent continually trained model from overfitting to limited examples in the buffer?
- Can authors provide any explanation why Lipschitz constant is higher in a continual-learning setup with buffered examples? I intuitively understand why is it so, but how does it relate theoretically?

- Not really a question, but an idea. It would be interesting for the future work to see if proposed solution might be used as a regularisation for generative rehearsal approaches in order to reduce overfitting to part of the data distribution that is better covered by a generative model.


**Limitations:**

Yes

**Strengths And Weaknesses:**

Strengths:
- Interesting initial insights and convincing motivation of the work
- Well prepared experimentation section wit setups that provide better understanding of what is the main impact of the proposed solution. Good performance evaluation with well selected baseline methods that tackle the problem of buffer based continual learning in different ways
- Experiments indicate that solution improves recent state-of-the-art approaches, for some setups by a large margin

Weaknesses:
- Comparison to related works is very limited, a few crucial state-of-the-art methods are missing in both related works and experiments. Most importantly similar work - "Gradient-based Editing of Memory Examples for Online Task-free Continual Learning" where buffer examples are adjusted in order to similarly to this method improve the performance of several baseline buffer based continual learning methods.
- In Eq. 5  an observation is introduced that ReLU can be understood as matrix multiplication by a diagonal matrix with either zeros or ones. On top of this an upper bound of Lipschitz constant for ReLU activation is proposed that can be interpreted as switching ReLU to identity function. This is a very significant simplification that basically reduces calculation of proposed Lipschitz based regularisation for all complex architectures as if they were linear models.
- For many experiments (iCARL - splitCUB, GDumb (smaller buffer), ER-ACE(pre-trained CIFAR100), Hessian Eigenvalues),  gain observed with proposed solution is marginal. Without confidence interval it is hard to evaluate if LIDER improves results over the random factor.


Small details:
- [116] also known
- [247] Why Unexpectedly? I guess it is rather expected that performance decreases when probability of switching labels in buffer poisoning increases.

---

> ### Author Response · Authors · 2022-08-02
> **Response to reviewer PLZ4 - Part 1/2**
>
> ## Weakness - Comparison to related works is very limited
> Following the insightful suggestions provided by PLZ4 and oByL, we are going to strengthen the experimental findings by extending both the related works and Sec.4 with new competitors, such as GMED [a] (pointed out by PLZ4) and several approaches mixing regularization and replay. Regarding GMED, we remark that it was conceived to work in the online setting, which differs from our setting where - in line with [b,c,d] - multiple epochs per task are allowed.
>
> Nevertheless, we provide the comparison w.r.t. GMED in both settings. For the single-epoch (online) scenario proper of GMED, we refer PLZ4 to the response we have provided to tK25 - *Experiments in the online setting*. Instead, the comparison in the multi-epoch scenario, please refer to the response to oByL - *Lack of competitors*. Unsurprisingly, GMED's update strategy based on gradient ascent is very effective in its original setting, but not as rewarding in our experimental setting.
>
> ## Weakness - Too simplistic an approximation
> We understand the reviewer's concern; the coarse approximation we employed might be further refined. Nevertheless, we acknowledge that it also allows for a computationally efficient and scalable proposal, suitable for the classical requirements of the continual learning scenario. In the following, we briefly recap the notions and tools through which our approximation can be favorably refined.
>
> In more details, it could help to provide, for each non-linear activation function $\sigma^{l}$, tighter upper bounds of its Lipschitz constant $||\sigma^{l}||_L$. We would like to clarify that our assumption (line 123 of the main paper - 133 in current revision) $||\sigma^{l}||_L \leq 1$ refers to the global $1$-Lipschitz continuity, which requires the inequality to hold for all points from the input domain. Such a requirement can be too strict and not representative of DNNs, as the input of a layer does not distribute uniformly but is often constrained in a subspace, whose shape depends on the activations of the previous layers. For this reason, recent works [e,f] often refer to a different property  *i.e.*, the local Lipschitz continuity, which bounds output perturbation only for certain input regions. Notably, it has been proved to provide a tighter upper bound (see Theorem 1 of [f]), and hence a more effective regularizing signal. On the other hand, it often requires a higher computational overhead, which could clash with the incremental scenario subject of our work.
>
> We plan to investigate the efficiency/accuracy trade-off inherent these advanced estimations in future works. Concerning our submitted paper, we are extending the section regarding the limitations of our approach with these considerations.
>
> [a] Jin, X., Sadhu, A., Du, J., & Ren, X. (2021). Gradient-based Editing of Memory Examples for Online Task-free Continual Learning. NeurIPS.
> [b] Buzzega, P., Boschini, M., Porrello, A., Abati, D., & Calderara, S. (2020). Dark Experience for General Continual Learning: a Strong, Simple Baseline. NeurIPS.
> [c] Rebuffi, S. A., Kolesnikov, A., Sperl, G., & Lampert, C. H. (2017). iCaRL: Incremental classifier and representation learning. CVPR.
> [d] Wu, Y., Chen, Y., Wang, L., Ye, Y., Liu, Z., Guo, Y., & Fu, Y. (2019). Large scale incremental learning. CVPR.
> [e] Jordan, M., & Dimakis, A. G. (2020). Exactly computing the local Lipschitz constant of relu networks. NeurIPS.
> [f] Huang, Y., Zhang, H., Shi, Y., Kolter, J. Z., & Anandkumar, A. (2021). Training certifiably robust neural networks with efficient local Lipschitz bounds. NeurIPS.

---

> > ### Author Response · Authors · 2022-08-02
> > **Response to reviewer PLZ4 - Part 2/2**
> >
> > ## Weakness - Marginal performance gain in some experiments
> > To shed light on this matter, we have re-run the experiments indicated by the reviewers ten times each, using the same hyperparameter configurations found in the suppl. materials. In the table below, we focus on the Final Average Accuracy (FAA) and report several statistical indicators,  *i.e.*, the top-1 value, the average, the standard error (SE), and the 25th-, 50th-, and 75th-percentiles (Q1, Q2, and Q3 respectively). Analyzing these results, it emerges that the gains of LiDER are consistent and not negligible even for those settings highlighted by the reviewer.
> >
> > |                   |Dataset                          |Buffer size             | Top-1 FAA   | avg. FAA     |SE         |Q1        |Q2        |Q3       |
> > |-------------------|---------------------------------|------------------------|-------------|---------------|-----------|----------|----------|---------|
> > |**ER-ACE**             |Split CIFAR-100 (w/ pret)       |2000   |$57.34$      |$56.89$          |$0.12$       |$56.51$     |$56.88$     |$57.28$    |
> > |**ER-ACE + LiDER**     |"       |"   |$\textbf{57.73}$  |$\textbf{57.40}$          |$0.05$       |$57.31$     |$57.35$     |$57.40$   |
> > |**GDumb**              |Split CUB-200 (w/ pret)         |400    |$09.47$      |$08.69$           |$0.14$       |$08.32$      |$08.54$      |$09.09$     |
> > |**GDumb + LiDER**      |"         |"    |$\textbf{10.51}$  |$\textbf{10.06}$          |$0.07$       |$09.89$      |$09.97$      |$10.21$     |
> > |**iCaRL**              |Split CUB-200 (w/ pret)         |1000   |$60.61$      |$59.99$          |$0.11$       |$59.72$     |$59.96$     |$60.20$   |
> > |**iCaRL + LiDER**      |"         |"   |$\textbf{61.21}$  |$\textbf{60.81}$          |$0.08$       |$60.62$     |$60.74$    |$61.02$     |
> >
> >
> > ## Question - Typo in line 247 (257 in current revision)
> > The presence of the word “Unexpectedly” is a typo: as PLZ4 rightly pointed out, the greater the poisoning injected in the buffer, the higher we expect the performance will be negatively affected. We are hence removing that word in the final version of the manuscript.
> >
> > ## Questions - Additional clarifications
> > To provide a better understanding of the value of lowering the Lipschitz constants in CL, we have modified the last paragraphs of the Introduction section, also following the hints of tK25 (who raised similar doubts). In particular, we have more clearly emphasized the issue of rehearsal strategies: namely, a low-data training regime leads to higher uncertainty, which shows as non-smooth decision boundaries. We then delve into the details of how Lipschitz-based regularization can come to the rescue by enforcing small variations against local input perturbations. For an additional depiction of such an intuition, we refer the reviewers to Fig.A of the [external .pdf](https://github.com/anonnips2022/LiDER/raw/master/extra_illustrations.pdf) containing the illustrations of the rebuttal. For further questions, please feel free to submit them to us in the rolling discussion.
> >
> > ## Question - Idea for future works
> > We thank the reviewer for their suggestion; actually, we have been applying the Lipschitz regularization in the past few months with the aim of enhancing the generative capabilities of diffusion models. We hope to submit this work to the scientific community as soon as possible.

---

### Official Review · Reviewer_QKp5 · 2022-07-07

**Rating:** 6
**Confidence:** 4
**Soundness:** 3 good
**Presentation:** 4 excellent
**Contribution:** 2 fair

**Summary:**

The authors focus on the Catastrophic Forgetting due to overfitting to the small dataset of memorised samples.
They show the degradation of the decision boundary due to overfitting to a subset of data on the first tasks.
They also show that the Lipchitz constant increases the smaller the memory buffer, which is tightly related to the decision boundary's deterioration.
They introduce a regularisation scheme to minimise the Lipchitz constant over the samples in the memory.

**Questions:**

- In 5.4, could you please explain the selection process of the fixed target ?
- Could you clarify which performance is reported in Table 4 ?

Tiny recommendations for clarity (don't impact the score) :
- In the appendix, presenting the hyper-parameters in a table format may make them more readable :)
- I think it would be more clear to add the standard errors in the same table as the means. Maybe split the tables into two tables : FAA and FF.

**Limitations:**

The authors have adequately addressed the limitations and potential negative impact of the work.

**Strengths And Weaknesses:**

## Originality
### 1- Are the tasks or methods new?

The method LiDER is new. However, the idea to constraint the Lipschitzness of the neural network and the way to do so is not new.

### 2- Is the work a novel combination of well-known techniques?

Yes, the work is an application of techniques to minimise the Lipschitzness of neural networks to Continual Learning.

### 3- Is it clear how this work differs from previous contributions?

Yes, the main way this work differs from previous contributions are. :
- First application of the Lipschitzness minimisation to Continual Learning
- The design of the optimisation objective on the Lipschitz constant for continual learning.

### 4- Is related work adequately cited ?
To my knowledge, the related work is adequately cited

## Quality
### 1- Is the submission technically sound ?

Yes, the submission is technically sound.

### 2- Are claims well supported (e.g., by theoretical analysis or experimental results) ?

Most claims were supported by experiments as evidence.

### 3- Are the methods used appropriate ?

The idea of focusing on the Lipschitzness seems relevant, given the improvements experimental results.
Also, several works have pointed to the link between the flatness of the loss and Catastrophic Forgetting.
Intuitively, a network with a smaller Lipschitzness constant would converge to a more stable optima.

### 4- Is this a complete piece of work or work in progress ?

This is a complete piece of work.

### 5- Are the authors careful and honest about evaluating both the strengths and weaknesses of their work ?

The authors discussed the limitations of their work in terms of :
- computational requirements.
- the non applicability of the method to attention based architectures.

## Clarity:
### 1- Is the submission clearly written?
- The paper is well written and easy to follow.
- The experimental setup is presented in detail
- The background is clearly presented and helped me ramp-up on the Lipschitzness of neural networks.

### 2- Is it well organised? (If not, please make constructive suggestions for improving its clarity.)

Overall the paper is well organised and was easy to follow.

### 3- Does it adequately inform the reader? (Note that a superbly written paper provides enough information for an expert reader to reproduce its results.)

The hyperparameters to reproduce the experiments are provided in the appendix. Also, the code was provided in supplementary materials.

## Significance:
### 1- Are the results important?

Strengths :
- The method is easy to implement and applicable as an addition to a large number of algorithms.
- The method achieves improvements on the order of 5 to 8% gains in FAA for split-ImageNet and Split CUB-200, for  DER++, and up to 10% in AF.

Weaknesses :
- The gains LiDER bring are most significant, mostly in split miniImageNet and Split CUB-200 and for a subset of the algorithms.

### 2- Are others (researchers or practitioners) likely to use the ideas or build on them?
- Given that the method is easy to implement, practitioners are likely to benchmark the idea against other methods for applications.
- However, it's unclear to which extend the computational overhead of the power iteration method makes LiDER applicable to large scale real world applications.

### 3- Does the submission address a difficult task in a better way than previous work?

The method is designed to be plugged on top of other methods. In the experiments, this leads to an improvement or the same performance in most cases.

### 4- Does it advance the state of the art in a demonstrable way?
- The method achieves improvements on the order of 5 to 8% gains in FAA for split-ImageNet and Split CUB-200, for  DER++, and up to 10% in AF.
- The gains LiDER bring are most significant, mostly in split miniImageNet and Split CUB-200 and for a subset of the algorithms.

---

> ### Author Response · Authors · 2022-08-02
> **Response to reviewer QKp5**
>
> ## Weakness - reported in the Significance sec. of the review
> We have some trouble understanding the reviewer's concern: the sentence "*The gains LiDER bring are most significant, mostly in Split miniImageNet and Split CUB-200 and for a subset of the algorithms.*" sounds more as a good point rather than a weakness. So, we kindly invite QKp5 to explain again its concern: in the case it regarded the marginal gain obtained for some experiments, we refer them to the response we have already provided to PLZ4, which addresses that issue.
>
> ## Question - Applicability to real world scenarios
> To investigate such an interesting point, we have benchmarked the *memory footprint/accuracy* and *speed/accuracy* trade-offs of our approach in combination with ER-ACE, for increasing complex backbone networks (ranging from EffecientNET-B0 to the deepest EffecientNET-B7). The experimental evaluation - whose results are shown in Fig.B of the [attached .pdf](https://github.com/anonnips2022/LiDER/raw/master/extra_illustrations.pdf) - has been carried out on Split CUB-200, the dataset with the highest input resolution in our tests. For a meaningful term of comparison, we also report the performance trend of ER-ACE without LiDER.
>
> As can be observed, our approach clearly involves an overhead, but this is fully rewarded by superior accuracy, especially for smaller architectures (EfficientNet-B≤4). As a final note, while our use of power iteration might seem a major hindrance to scalability, we observed in practice that few iterations were enough to obtain good and stable estimates of the eigenvalues.
>
> ## Question - Sec 5.4 and the selection process of the Lipschitz targets
> Sec 5.4 reported that our strategy - which lets the optimization find the best values of the bounds $c_k$ - performs better than simply fixing constant target values prior to training. To support our claim, we used the grid search strategy for comparison, running the experiments with different values of $c_k \equiv c \in (0.0, 0.1, 0.5)$ and reporting the best result in Tab.4.
>
> ## Minor recommendation - Clarity
> We agree with the reviewer's suggestions: however, we had to focus on other issues during the restricted time allowed for the rebuttal; we will make sure to address them in the final version of the paper.

---

### Official Review · Reviewer_oByL · 2022-07-08

**Rating:** 6
**Confidence:** 4
**Soundness:** 3 good
**Presentation:** 3 good
**Contribution:** 3 good

**Summary:**

The paper proposes inclusion of a smoothness regulariser in rehearsal training for alleviating the problem of catastrophic forgetting.  Authors present empirical evidence showing that adding the proposed regularise to existing rehearsal techniques reduces forgetting.

**Questions:**

Why is the regularisation objective in equation 9 is so complicated?  In on term you try to make eigenvalue equal to c, in the other term you try to minimise c.  Why not just create an objective that minimises the eigenvalue?  Surely, if eigenvalue is less than c, that’s good...and so the first term is not always doing the right thing…and the combination of two is just a very complicated way of minimisation of the eigenvalue.

**Limitations:**

The limitation is that based on the lack of evaluation of the propose regularisation method against existing ones, it’s hard to tell if it is that good on its own.  As an investigation of the effectiveness of combining rehersal and regularisation, it’s not exhaustive enough, because it only takes into consideration a new regularistion method.  I think, as it is, the work is a good start to a possibly interesting investigation, that needs much more done in terms of critical evaluation of various combinations of rehearsal and other regularisation methods.

**Strengths And Weaknesses:**

This paper tackles two aspects of catastrophic forgetting: a new regularisation objective and the idea of combining regularisation (that alleviates catastrophic-forgetting) with rehearsal methods.  So I’ll tackle these two aspect separately.

As a new regulariser, I think the propose idea is not that fresh.  It seems like it’s just another facet of existing regularisation techniques that attempt to find flat minima for each of the tasks (Mirzadeh.etal 2020, Yin.etal 2021).   In fact, both those works already propose minimisation of the eigenvalue of the hessian matrix of the model - the only thing new here would be doing that on a per layer basis on weights…which may make the proposed method computational simpler.  Also, whereas Mirzadeh.etal2020 and Yin.etal2021 make  assumptions about Lipschitz smoothness of the learning model, in this work the objective is to maximise that smoothness.  I wonder if this work (just like Fisher-matrix based Elastic Weight Consolidation) is a special case of the Hessian-based regularisation proposed in Mirzadeh.etal2020 and Yin.etal2021?  If so, is the proposed estimator of smoothness better than the one introduced in Ritter.etal.2018?  Hard to tell at the moment, since the authors here don’t compare their regulariser against existing ones.

The second aspect, of combining rehearsal methods with regularisation methods for the purposes of reducing catastrophic forgetting is kind of interesting - perhaps a bit obvious, but nevertheless something that needs to be explored and evaluated.  However, in this case, I think there needs to be an empirical evaluation using other regularisers in addition to the proposed one.  Is the combination of rehearsal and the proposed regulariser better than a combination of rehearsal and the smart choices of training that flatten the minima proposed in Mirzadeh.etal2020?  Or is it better than combination of rehearsal than the regulariser of Ritter.etal.2018?  Is it slightly worse, but much faster?  And since these existing regularisers rely on Taylor series approximation of the Hessian of the loss function at minima of given tasks, could the fact that in rehersal we have access to some data from previous tasks be useful to improve that estimation?  Yin.etal2021 show the exact relationship of the error in the estimate of the Hessian to forgetting.



Mirzadeh.etal2020 - “Understanding hte role of training regimes in continual learning”
Yin.etal2021 - “Optimization and Generalization of Regularization-Based Continual Learning: a Loss Approximation Viewpoint”
Ritter.etal.2018 - “Online structured Laplace approximations for overcoming catastrophic forgetting”

---

> ### Author Response · Authors · 2022-08-02
> **Response to reviewer oByL - Part 1/3**
>
> ## Weakness - The idea is not fresh
> While QKp5 recognizes novelty as a strength of our work, oByL denounces its close similarity with those approaches that "*attempt to find flat minima for each of the tasks*", such as Mirzadeh et al. [a] and Yin et al. [b]. We suspect this is the outcome of a superficial analysis: **substantial differences** exist between our idea and the line pointed out by the reviewer. Indeed, these works pursue flatness of the loss landscape w.r.t. weights (*i.e.*, they reason in **parameter space**): in other words, they encourage the model to be robust when perturbations are applied to its **weights**. Differently, we seek models that are robust w.r.t. changes in **input space**.
>
> The two lines - which may share similar keywords or exploit the same mathematical tools, such as the Hessian and Lipschitz continuity - build upon orthogonal axes (weights vs input): therefore, it is fundamental to assess which of the two is used as reference domain. As an example, the authors of [b] actually involved Lipschitz continuity of the Hessian (see Definition 1 and 2 of that paper) as said by oByL, but they did it with regard to weights variations.
>
> Regarding the bridge between the two sides, it is an active and very recent area [c,d,e,f]. As reported in [c], there is theoretically no correlation between the Hessian w.r.t. weights and the robustness of the model w.r.t. the input: as depicted in Fig. 1 of [e], although the loss surface in the parameter space can show a flat minima, there is nothing that prevents non-smooth variations in the input space. However, the authors of [c] empirically found that models with higher Hessian spectrum w.r.t. weights are also more prone to adversarial attacks. A similar thesis has been argued by the authors of [e], while the third result reported in [f] seems to refute it. Furthermore, Sec. 5.3 of our paper investigated the opposite link and revealed that CL models trained to be robust w.r.t. input changes tend to attain flatter minima in parameter space. To sum up, bridging the gap between these two properties is still open to debate and worth-exploring; concerning our paper that focused on continual learning, we consider it precocious and slightly out of scope.
>
> ## Weakness - Lack of competitors
> Both oByL and PLZ4 stressed that our work could be strengthened by a more thorough evaluation. For this reason, we ran several new experiments encompassing three of the four settings of Tab. 1 & 2 of the main paper with the following competitors:
>
> + *Stable SGD (sSGD)* [a], which introduces some specific alterations to the model's training regime;
> + *online EwC (oEWC)* [g] and *Online Laplace (oLAP)* [h], which apply Hessian-based parameter-importance estimation and constrain the most significant model parameters for previous tasks;
> + *Gradient-based Memory Editings (GMED)* [i], a state-of-the-art replay method designed for the online (single-epoch) setting, which adjusts its buffer examples to mitigate forgetting.
>
> As one can observe from the results presented in the table below, all tested regularization methods yield lackluster results, even failing to outperform the fine-tuning lower bound in the more complex benchmarks. This is expected, as regularization methods are well-known to underperform in the CIL setting [j,k,l].
>
> | |Split CIFAR-100 | Split CIFAR-100 |Split miniImageNet | Split miniImageNet |Split CUB-200 | Split CUB-200 |
> |--|--|--|--|--|--|--|
> |Method |*w/o pre-training*| *w/o pre-training* |*w/o pre-training*| *w/o pre-training* |*w/ pre-training* | *w/ pre-training*|
> ||||||||
> |**sSGD** |$09.43~(89.58)$ | |$04.49~(78.07)$ | |$05.84~(51.42)$ ||
> |**oEwC** |$09.16~(84.48)$ | |$03.01~(49.99)$ | |$08.70~(83.00)$ ||
> |**oLAP** |$09.37~(88.69)$ | |$04.04~(73.25)$ | |$08.22~(79.01)$ ||
> |Buffer Size|*500* |*2000*|*2000*|*5000*|*400* |*1000*|
> |**GMED** |$20.49~(70.51)$ |$29.40~(60.99)$ |$09.82~(61.77)$ |$16.18~(48.99)$ |$11.39~(68.29)$ |$18.87~(64.45)$ |
>
> While more effective, GMED shows much less reliable w.r.t. all other tested rehearsal approaches. We attribute this to the fact that GMED is designed for and tested on a single-epoch online CL evaluation paradigm. Instead, when we give the learner plenty of steps to fit each task, we conjecture that GMED's buffer modification strategy turns from an anchor against forgetting into a self-destructive liability, as it increasingly alters replay examples. We remark that such an issue does not show in the online setting, as testified by the comparison provided to tK25.

---

> > ### Author Response · Authors · 2022-08-02
> > **Response to reviewer oByL - Part 2/3**
> >
> > We examine another point raised by oByL and investigate how coupling the above-mentioned regularization techniques with replay strategies (such as ER-ACE and DER++) compares with our solution. We present the results in the following tables.
> >
> > ||Split CIFAR-100|Split CIFAR-100 |Split miniImageNet|Split miniImageNet |Split CUB-200 | Split CUB-200 |
> > |-|-|-|-|-|-|-|
> > |Method|*w/o pre-tr.* |*w/o pre-tr.* |*w/o pre-tr.*|*w/o pre-tr.*|*w/ pre-tr.*|*w/ pre-tr.*|
> > |Buffer Size|500 |2000|2000|5000|400 |1000|
> > |**ER-ACE** |$36.48~(38.21)$ |$48.41~(27.90)$ |$22.60~(23.74)$ |$27.92~(19.72)$ |$41.83~(26.42)$ |$51.98~(18.79)$ |
> > |**ER-ACE + LiDER** |$38.43~(36.00)$ |$\textbf{50.32}~(28.30)$|$24.13~(25.97)$ |$\textbf{30.00}~(19.99)$|$\textbf{50.89}~(20.79)$|$\textbf{60.92}~(14.62)$|
> > ||||||||
> > |**ER-ACE + sSGD**|$\textbf{39.59}~(34.29)$|$49.70~(24.44)$ |$22.43~(13.99)$ |$24.12~(11.02)$ |$22.67~(21.18)$ |$29.88~(14.24)$ |
> > |**ER-ACE + oEwC**|$35.06~(38.08)$ |$45.59~(27.55)$ |$\textbf{24.32}~(27.09)$|$29.46~(20.11)$ |$48.34~(27.59)$ |$59.74~(17.36)$ |
> > |**ER-ACE + oLAP**|$36.58~(37.88)$ |$47.66~(29.34)$ |$23.19~(28.69)$ |$28.77~(21.85)$ |$42.64~(29.57)$ |$52.86~(19.24)$ |
> >
> > ||Split CIFAR-100|Split CIFAR-100 |Split miniImageNet|Split miniImageNet |Split CUB-200 | Split CUB-200 |
> > |-|-|-|-|-|-|-|
> > |Method|*w/o pre-tr.* |*w/o pre-tr.* |*w/o pre-tr.*|*w/o pre-tr.*|*w/ pre-tr.*|*w/ pre-tr.*|
> > |Buffer Size|500 |2000|2000|5000|400 |1000|
> > |**DER++**|$37.13~(49.80)$ |$52.08~(31.10)$ |$23.44~(46.69)$ |$30.43~(37.11)$ |$49.30~(36.05)$ |$61.42~(19.95)$ |
> > |**DER++ + LiDER**|$\textbf{39.25}~(45.50)$|$\textbf{53.27}~(27.51)$|$\textbf{28.33}~(36.29)$|$\textbf{35.04}~(25.02)$|$\textbf{57.90}~(27.55)$|$\textbf{67.97}~(14.44)$|
> > ||||||||
> > |**DER++ + sSGD** |$38.48~(39.70)$ |$50.74~(25.56)$ |$19.29~(18.73)$ |$24.24~(28.16)$ |$31.08~(30.44)$ |$41.69~(21.96)$ |
> > |**DER++ + oEwC** |$35.22~(52.13)$ |$51.53~(32.18)$ |$24.53~(47.90)$ |$31.91~(36.35)$ |$51.86~(30.14)$ |$62.54~(16.90)$ |
> > |**DER++ + oLAP** |$34.48~(55.84)$ |$50.80~(35.14)$ |$25.02~(40.65)$ |$32.78~(32.92)$ |$49.56~(32.68)$ |$63.27~(15.90)$ |
> >
> > We observe that sSGD improves ER-ACE and DER++ only on the simpler Split CIFAR-100 benchmark, whereas its performance degrades severely both on the more complex Split *mini*ImageNet and on Split CUB-200, where we suspect it fails to effectively exploit the pre-trained network. By contrast, we observe satisfactory improvements from the application of oEwC and oLAP, which become particularly relevant in the presence of pre-training. We can explain this finding by considering that pre-training has a known flattening effect on the loss landscape [m]; in such a setting, encouraging the model to stay close to its prior (as done by these methods) is especially rewarding.
> >
> > All things considered, we observe that our proposed LiDER proves almost always more effective than any of the other tested approaches. Again, we argue that our proposal lies on an orthogonal axis w.r.t.  typical CL parameter regularization; on the basis of these results, we believe that this difference emerges also experimentally.
> >
> > Still, we agree with oByL: combining parameter regularization approaches and rehearsal represents a very interesting new research direction.  However, we argue that regularization approaches acting in parameter space require tailored modifications to profit from the replay memory, which we deem out of scope for this work. As a small proof of concept, we slightly altered the way the Fisher Information Matrix is estimated in oEwC at the end of each task, by computing it on the i.i.d. memory buffer in lieu of the last encountered dataset. As we show in the results below, this simple prototype (indicated with $F$ on $\mathcal{M}$) further boosts the performance of oEwC, confirming that this should be the focus of future endeavors in the CL community.
> >
> > ||Split CIFAR-100| Split CIFAR-100 |Split miniImageNet|Split miniImageNet |Split CUB-200 | Split CUB-200|
> > |-|-|-|-|-|-|-|
> > |Method|*w/o pre-tr.*|*w/o pre-tr.*|*w/o pre-tr.*|*w/o pre-tr.* |*w/ pre-tr.* |*w/ pre-tr.*|
> > | Buffer Size|*500* |*2000*|*2000*|*5000*|*400* |*1000*|
> > | **ER-ACE + oEwC**|$35.06~(38.08)$ |$45.59~(27.55)$ |$24.32~(27.09)$ |$29.46~(20.11)$ |$48.34~(27.59)$ |$59.74~(17.36)$ |
> > | **ER-ACE + oEwC + $F$ on $\mathcal{M}$**|$\textbf{37.70}~(35.51)$|$\textbf{49.26}~(24.53)$|$\textbf{24.55}~(26.80)$|$\textbf{29.69}~(18.93)$|$\textbf{48.77}~(26.82)$|$\textbf{59.96}~(16.18)$|
> > ||||||||
> > | **DER++ + oEwC** |$35.22~(52.13)$ |$\textbf{51.53}~(32.18)$|$24.53~(47.90)$ |$31.91~(36.35)$ |$51.86~(30.14)$ |$62.54~(16.90)$ |
> > | **DER++ + oEwC + $F$ on $\mathcal{M}$**|$\textbf{35.45}~(52.76)$|$51.10~(32.79)$ |$\textbf{25.86}~(47.11)$|$\textbf{32.00}~(37.43)$|$\textbf{56.41}~(27.45)$|$\textbf{67.02}~(14.80)$|
> >
> > When revising our paper, we will include the most significant new experimental results in the main paper and report the extended discussion on regularization and replay in the supplementary materials.

---

> > > ### Author Response · Authors · 2022-08-02
> > > **Response to reviewer oByL - Part 3/3**
> > >
> > > ## Question - Eq. 9
> > > The aim of the twofold objective is to avoid degenerate solutions. Indeed, when optimizing only the first term (Eq. 7), the $c_k$ variables subject to direct optimization end up tracking the $\lambda_1^k$ values. On the other hand, the optimization of Eq. 8 without proper counterbalancing could over-regularize the model and lead to insensitive behavior. For an empirical assessment of the values of both objectives, we refer the reviewer to the response provided to tK25 regarding the sensitivity of performance w.r.t. the corresponding balancing coefficients.
> > >
> > > [a] Mirzadeh, S. I., Farajtabar, M., Pascanu, R., & Ghasemzadeh, H. (2020). Understanding the role of training regimes in continual learning. NeurIPS.
> > > [b] Yin, D., Farajtabar, M., Li, A., Levine, N., & Mott, A. (2020). Optimization and generalization of regularization-based continual learning: a loss approximation viewpoint. arXiv.
> > > [c] Yao, Z., Gholami, A., Lei, Q., Keutzer, K., & Mahoney, M. W. (2018). Hessian-based analysis of large batch training and robustness to adversaries. NeurIPS.
> > > [d] Stutz, D., Hein, M., & Schiele, B. (2021). Relating adversarially robust generalization to flat minima. ICCV.
> > > [e] Yu, F., Qin, Z., Liu, C., Zhao, L., Wang, Y., & Chen, X. (2019). Interpreting and evaluating neural network robustness. IJCAI.
> > > [f] Kamath, S., Despande, A., & Subrahmanyam, K. V. (2019). On Adversarial Robustness of Small vs Large Batch Training. ICML Workshop.
> > > [g] Schwarz, J., Czarnecki, W., Luketina, J., Grabska-Barwinska, A., Teh, Y. W., Pascanu, R., & Hadsell, R. (2018). Progress & compress: A scalable framework for continual learning. ICML.
> > > [h] Ritter, H., Botev, A., & Barber, D. (2018). Online structured Laplace approximations for overcoming catastrophic forgetting. NeurIPS.
> > > [i] Jin, X., Sadhu, A., Du, J., & Ren, X. (2021). Gradient-based Editing of Memory Examples for Online Task-free Continual Learning. NeurIPS.
> > > [j] Farquhar, S., & Gal, Y. (2018). Towards Robust Evaluations of Continual Learning. ICML Workshop.
> > > [k] Buzzega, P., Boschini, M., Porrello, A., Abati, D., & Calderara, S. (2020). Dark Experience for General Continual Learning: a Strong, Simple Baseline. NeurIPS.
> > > [l] Aljundi, R., Lin, M., Goujaud, B., & Bengio, Y. (2019). Gradient based sample selection for online continual learning. NeurIPS.
> > > [m] Mehta, S. V., Patil, D., Chandar, S., & Strubell, E. (2021). An empirical investigation of the role of pre-training in lifelong learning. ICML.

---

> > > > ### Comment · Reviewer_oByL · 2022-08-08
> > > > **Thanks for detailed reply and extra work**
> > > >
> > > > I think the point about the work I referenced being concerned with perturbation of the model weights vs. this work's concern with the perturbations of the data in the input space is well made.  The tables in Part 2 of the reply are also very informative and constitute strong evidence that the propose method brings something new to the table.  I withdraw my reservations of originality and commend the authors on making the extra effort of generating all these additional results, which I think improve the paper significantly.  I am happy to raise may review rating all the way to "weak accept".  The reason why I am not rating it higher is that there is something that bothers me about the results provided in the part 1 of the reply.  Really, all those other methods perform so poor?  It's just doesn't seem right... and I struggle to understand why this would be the case.  Why these, supposedly state of the art, methods are doing so poorly?  Seems like perhaps there is something about the experiment generating those results that makes it an unfair comparison...

---

> > > > > ### Author Response · Authors · 2022-08-09
> > > > > **Thanks and additional clarification**
> > > > >
> > > > > We are truly thankful to oByL for taking the time to read through our response and rethink their initial evaluation.
> > > > >
> > > > > Regarding the concern raised about the performance of competitors in our response, we remark that we follow a Class-Incremental (CIL) evaluation protocol, in which all evaluated tasks share the same classifier. By contrast, the suggested parameter regularization-based competitors refer to the simpler Task-Incremental (TIL - separate classifier for each task) or Domain-Incremental (DIL - shared classifier, but all classes are present in all tasks, with only changes in their input distributions) protocols.
> > > > >
> > > > > Several works in recent literature recognize that the CIL protocol (a.k.a. shared head) we chose is the most realistic and challenging evaluation protocol [a, b, c, d, e]. If one considers works that - like ours - target CIL and use non-trivial memory buffer sizes, it can be seen that weight-regularization based methods perform poorly, in line with what is reported in our table [e, a, d, f (tables B1 and B2)].
> > > > >
> > > > > [a] Hsu, Y. C., Liu, Y. C., Ramasamy, A., & Kira, Z. (2018). Re-evaluating continual learning scenarios: A categorization and case for strong baselines. NeurIPS Workshop.
> > > > > [b] Aljundi, R., Lin, M., Goujaud, B., & Bengio, Y. (2019). Gradient based sample selection for online continual learning. NeurIPS.
> > > > > [c] Aljundi, R., Caccia, L., Belilovsky, E., Caccia, M., Lin, M., Charlin, L., & Tuytelaars, T. (2019). Online Continual Learning with Maximally Interfered Retrieval. NeurIPS.
> > > > > [d] Buzzega, P., Boschini, M., Porrello, A., Abati, D., & Calderara, S. (2020). Dark experience for general continual learning: a strong, simple baseline. NeurIPS.
> > > > > [e] Farquhar, S., & Gal, Y. (2018). Towards robust evaluations of continual learning. ICML Workshop.
> > > > > [f] Prabhu, A., Torr, P. H., & Dokania, P. K. (2020). Gdumb: A simple approach that questions our progress in continual learning. ECCV.

---

### Official Review · Reviewer_tK25 · 2022-07-15

**Rating:** 6
**Confidence:** 3
**Soundness:** 3 good
**Presentation:** 2 fair
**Contribution:** 3 good

**Summary:**

The authors propose to constrain lipschitz constant  for the Continual Learning settings in order to control problems related to reusing the same replay samples. They show that this leads to increased performance across a variety of rehearsal based methods

**Questions:**

- The method introduces additional hyperparameters alpha beta, how robust are these?
- What would happen if the incoming data, not the buffer data was used to compute  L_c-Lip?

**Limitations:**

Yes

**Strengths And Weaknesses:**

*Strengths*

- The method performs well across a variety of datasets and methods.



*Weakness*
- the motivation can be hard to follow at times, particularly the 3rd to last paragraph of the introduction is  vague and overly informal. Figure 1 should be more clearly explained as it takes sometime to understand what it is trying to show
- The authors do not consider the online setting where rehearsal methods are commonly applied

---

> ### Author Response · Authors · 2022-08-02
> **Response to reviewer tK25 - Part 1/2**
>
> ## Weakness - Clarity
> We modified the parts of the main paper subject to the concern of the reviewer: namely, the last paragraphs of the Introduction and the explanation of Fig.1. We hope that these amendments can further help clarify the motivations of our work, which were welcomed by reviewers PLZ4 and QKp5.
>
> ## Weakness - Experiments in the online setting
> The investigation of the online setting is certainly worth-noting, on par with the one we used. However, we advocate for what has been said in [a]: when only one epoch is allowed on the current task, even the pure-SGD baseline fails at fitting it with adequate accuracy, especially with complex datasets such as CIFAR-100 and *mini*ImageNet. Therefore, the resulting performance - and in turn the comparisons among different approaches - can be difficult to read, as the effects of catastrophic forgetting and those linked to underfitting interleave here. Furthermore, it is even more complicated comparing approaches that were conceived only in either of the two settings (multi-epochs *vs.* single-epoch) such as ours and GMED [b].
>
> We therefore exhort the reader to interpret cautiously the results provided in the table below, reporting the single-epoch performance of various methods equipped with our regularizer.
>
> |Split CIFAR-100        |20 Tasks w/o pret.                  |20 Tasks w/o pret.                  |10 Tasks w/o pret.                  |10 Tasks w/o pret.       |
> |------------|------------------|------------------|------------------|-------------|
> |*FAA* (*FF*)           |$\|\mathcal{M}\|$=500               |$\|\mathcal{M}\|$=2000              |$\|\mathcal{M}\|$=500               |$\|\mathcal{M}\|$=2000   |
> ||||||
> |**GMED**               |$\textbf{22.39}~(66.89)$	                 |$\textbf{36.18}~(49.97)$                   |$\textbf{23.69}~(58.22)$                   |$\textbf{34.07}~(45.47)$        |
> |**DER++**              |$14.03~(51.61)$                       |$19.33~(42.43)$                       |$10.49~(41.46)$                       |$23.56~(26.73)$	         |
> |**+ LiDER**            |$\underline{15.54}~(50.34)$           |$\underline{21.93}~(44.78)$           |$\underline{14.65}~(43.58)$           |$\underline{24.36}~(28.50)$|
> |**ER-ACE**             |$17.58~(11.79)$                       |$21.60~(10.17)$                       |$17.23~(10.24)$                       |$21.73~(05.69)$            |
> |**+ LiDER**            |$\underline{18.28}~(11.66)$	         |$\underline{25.19}~(09.78)$           |$\underline{18.98}~(09.56)$           |$\underline{24.86}~(03.83)$|
>
> At a first glance, the results provided herein show a remarkable improvement for both DER++ and ER-ACE when equipped with LiDER, with a respective average gain of 2.27% and 2.29%. While consistent, this is not sufficient to make the baseline methods competitive against a method specifically designed for the online setting, such as GMED.
>
> ## Question - Robustness to hyperparameters $\alpha$ and $\beta$
> Fig.C in the [attached .pdf](https://github.com/anonnips2022/LiDER/raw/master/extra_illustrations.pdf) proposes a 2D summary of the performance variation yielded by different values of $\alpha$ and $\beta$ (introduced by Eq.9 of the main paper). In particular, each item of these matrices reports, for a given combination, the difference w.r.t. the average measured FAA. As can be observed, we distinguish two separate regimes: one for the cases when models are trained from scratch; another when using a pre-trained model.
>
> Without pre-training, we obtain a performance gain if the two parameters are comparable, with $\beta \geq \alpha$. If $\alpha > \beta$, we are overemphasizing the contribution of the first term of Eq.9 (which brings each layer's $\lambda^{k}_1$ and $c_k$ close to each other) over the second one (which induces small Lipschitz targets). For a randomly initialized model, this may lead the initial value of $\lambda^{k}_1$ to mislead $c_k$. Differently, if $\beta \gg \alpha$, the model is encouraged to over-regularize its response, resulting in a reduced capability of fitting the encountered data.
>
> Instead, if the backbone is pre-trained, we see an overall much stabler behavior, due to the $\lambda^{k}_1$ for each layer being well-behaved from the beginning of training. Here, the only pitfall is given by $\beta \gg \alpha$, which again leads the model to oversmoothing.
>
> [a] Buzzega, P., Boschini, M., Porrello, A., Abati, D., & Calderara, S. (2020). Dark Experience for General Continual Learning: a Strong, Simple Baseline. NeurIPS.
> [b] Aljundi, R., Lin, M., Goujaud, B., & Bengio, Y. (2019). Gradient based sample selection for online continual learning. NeurIPS.

---

> > ### Author Response · Authors · 2022-08-02
> > **Response to reviewer tK25 - Part 2/2**
> >
> > ## Question - Applying LiDER only on incoming datapoints
> > Following the hint of the reviewer, we have performed several experiments switching the target of regularization: not the examples from the replay memory, but instead those belonging to the current task. The tables below report the results obtained by the two strategies: while the original approach delivers consistent improvements across most of the settings, applying LiDER on incoming examples delivers inferior results.
> >
> > We believe that this evidence can be explained by examining the distinct data regimes for the current and previous tasks in continual learning. While learning the current task, indeed, the model can access many and many samples from its underlying distribution; therefore, the epistemic uncertainty [c] reduces and the learned decision boundaries are likely to be smoother. In this case, the Lipschitz regularization could represent an overkill, threatening to restrain the learning with no advantages. In our formulation, instead, only few examples are available for retaining the knowledge of past tasks: the risk of progressive overfitting - which we expressed through the progressive degradation of decision boundaries - is more severe: therefore, tailored countermeasures are more likely to be effective.
> >
> > ||CIF100 (w/o pt)|CIF100 (w/o pt)| CIF100 (w/ pt)  |CIF100 (w/ pt)     |
> > |---|---|---|---|---|
> > |Buffer Size|500                              |2000                             |500                         |2000 |
> > ||||||
> > |**DER++ + LiDER**             |$\textbf{39.25}~(45.50)$           |$\textbf{53.27}~(27.51)$           |$\textbf{45.37}~(48.16)$      |$\textbf{60.88}~(25.16)$      |
> > |**DER++ + LiDER (stream)**     |$34.78~(54.41)$	                 |$49.76~(35.91)$                    |$44.48~(49.11)$               |$59.39~(28.94)$	  |
> > |**ER-ACE + LiDER**            |$\textbf{38.43}~(36.00)$	         |$50.32~(28.30)$                    |$\textbf{48.97}~(28.58)$      |$\textbf{57.39}~(25.37)$	   |
> > |**ER-ACE + LiDER (stream)**   |$37.54~(36.46)$	                 |$\textbf{50.37}~(27.24)$           |$48.94~(28.80)$	            |$57.07~(23.50)$      |
> >
> > ||*mini*ImageNet (w/o pt)|*mini*ImageNet (w/o pt)     |CUB200 (w/ pt)     |CUB200 (w/ pt)     |
> > |---|---|---|---|---|
> > |Buffer Size|2000                        |5000                     |400                     |1000 |
> > ||||||
> > |**DER++ + LiDER**             |$\textbf{28.33}~(36.29)$      |$\textbf{35.04}~(25.02)$ |   $\textbf{57.90}~(27.55)$  |$\textbf{67.97}~(14.44)$	   |
> > |**DER++ + LiDER (stream)**    |$24.84~(44.54)$	            |$31.05~(34.85)$            |$56.96~(28.42)$           |$66.63~(16.30)$	  |
> > |**ER-ACE + LiDER**            |$\textbf{24.13}~(25.97)$      |$\textbf{30.00}~(19.99)$   |$\textbf{50.89}~(20.79)$  |$\textbf{60.92}~(14.62)$      |
> > |**ER-ACE + LiDER (stream)**   |$23.35~(23.56)$	            |$29.25~(18.78)$            |$48.44~(25.16)$	       |$59.60~(15.21)$      |
> >
> >
> > [c] Kendall, A., & Gal, Y. (2017). What uncertainties do we need in bayesian deep learning for computer vision?. NeurIPS.

---

### Author Response · Authors · 2022-08-02
**Summary of Author Feedback**

We would like to thank the reviewers for the time they spent and for their insightful suggestions. In the following, we propose a brief summary of the main points covered in our response.


+ **Weakness - Comparison to related works, issued by PLZ4 and oByL:** We have extended the experimental comparisons following their suggestions.
+ **Weakness - Too simplistic an approximation, issued by PLZ4:** We have provided a deep discussion on how our approximation can be further improved.
+ **Weakness - Marginal performance gain, issued by PLZ4:** We have repeated ten times the experimental settings they pointed out: by doing so, the improvements due to our approach emerge clearly.
+ **Weakness - The method is not fresh, issued by PLZ4:** We have clarified the deep distinction between our regularization term (acting in input space) and the works pointed out by the reviewer (acting in parameter space).
+ **Weakness - Clarity, issued by tK25:** We have modified the Introduction section to meet their concerns and make it easier to follow our motivations. The new version is highlighted in red.
+ **Weakness - Experiments in the online setting, issued by tK25:** We have benchmarked our approach in the single-epoch scenario, finding that it still provides improvements.

We produced a revised version of our original manuscript, where changed parts are highlighted in red.
As our response involves new illustrations, we made them available to reviewers at the [following link](https://github.com/anonnips2022/LiDER/raw/master/extra_illustrations.pdf). Furthermore, based on reviewers' questions and suggestions, we have also provided new insights and experimental investigations, covering applicability to large-scale scenarios, sensitivity w.r.t. hyperparameters, etc. We would like to remark that all of the reported results are reproducible; the code repository is available at the [following link](https://github.com/anonnips2022/LiDER/).

We are going to include these new concepts in both the main paper and the supplementary materials after the rebuttal period.

---

### Meta-Review · Area_Chair_Z6iW · 2022-08-24

**Recommendation:** Accept
**Confidence:** Certain

**Metareview:**

In order to solve the inherent problem of rehearsal-based continual learning methods (a problem in which a small pool of data from the previous tasks is repeatedly used for learning and hence we have a tight and unstable decision boundary), this paper proposes a method to provide smoothness of the backbone network by placing constraints on the Lipschitz constant of each layer. All reviewers unanimously recognized the strengths of this paper - sufficient performance improvement  in the designed experiments and convincing insight/motivation etc. However, some reviewers were concerned that the baselines or experimental settings are somehow weak, and the authors partially resolved this through additional experiments during discussion phase. Nevertheless, more diverse experimental results need to be added in the final version. Especially, this ac thinks that if the experimental results for more various buffer sizes are included in the final version, the experimental quality will be improved and the insight of the authors can be better supported.


**Award:**

No

---

### Decision · Program_Chairs · 2022-09-14

Accept